# Hybrid Boundary Physics-Informed Neural Networks for Solving Navier-Stokes Equations with Complex Boundary

Chuyu Zhou[*1], Tianyu Li[*2], Chenxi Lan[2], Rongyu Du[4], Guoguo Xin[1,+], Hangzhou Yang[1], Guoqing Wang[2,++], Xun Liu[3], Wei Li[3,+++]

[1]School of Physics, Northwest University, Xi'an 710127, China
[2]School of Computer Science and Engineering, University of Electronic Science and Technology of China, Chengdu 611731, China
[3]Department of Imaging Technology, Beijing Institute of Space Mechanics and Electricity, Beijing 100094, China
[4]School of Information Science & Technology, Northwest University, Xi'an 710127, China
[5]Donghai Laboratory, Zhoushan, Zhejiang 316000, China
[+]xinguo@nwu.edu.cn
[++]qwang0420@hotmail.com
[+++]wei_li_bj@163.com

## Abstract

Physics-informed neural networks (PINN) have achieved notable success in solving partial differential equations (PDE), yet solving the Navier-Stokes equations (NSE) with complex boundary conditions remains a challenging task. In this paper, we introduce a novel Hybrid Boundary PINN (HB-PINN) method that combines a pretrained network for efficient initialization with a boundary-constrained mechanism. The HB-PINN method features a primary network focused on inner domain points and a distance metric network that enhances predictions at the boundaries, ensuring accurate solutions for both boundary and interior regions. Comprehensive experiments have been conducted on the NSE under complex boundary conditions, including the 2D cylinder wake flow and the 2D blocked cavity flow with a segmented inlet. The proposed method achieves state-of-the-art (SOTA) performance on these benchmark scenarios, demonstrating significantly improved accuracy over existing PINN-based approaches.

## 1 Introduction

Fluid mechanics is an important field in science and engineering that deals with the study of the motion of liquids and gases. The NSE are basic partial differential equations that describe the dynamic behavior of viscous fluids, which are highly nonlinear partial differential equations and are widely used in aerodynamics, meteorology, oceanography, and industrial process simulations Munson et al. [1]. Although CFD methods have achieved considerable maturity, challenges such as mesh generation persist and can lead to numerical instability and reduced accuracy[2]. In recent years, PINNs, which integrate physical prior knowledge with deep learning [3], have emerged as a notable class of surrogate models for computational fluid dynamics.

PINN represent a deep learning methodology that incorporates physical constraints by embedding the governing physical equations and boundary conditions into the neural network's loss function. First proposed by Raissi et al. in 2019 [4], PINN eliminate the need for mesh generation, a requirement

inherent to traditional CFD solvers, while enabling more efficient resolution of inverse problems through data integration.

However, for fluid models with complex boundary conditions, conventional PINN methods often struggle to accurately approximate both the boundary conditions and the partial differential equations [5]. For conventional PINN, boundary conditions and initial conditions are explicitly embedded into the loss function, which is trained simultaneously with the PDE loss [4, 6]. A notable limitation of this loss formulation lies in its frequent failure to ensure simultaneous minimization of the PDE residual loss and the boundary condition loss. This issue becomes particularly pronounced when boundary conditions exhibit high complexity, thereby compromising the accuracy of the resultant solutions [7, 8, 9].

Here, to address the difficulty in balancing boundary condition losses and equation losses in complex boundary flow field problems, we propose a composite network approach specifically for complex boundary issues, named HB-PINN. By independently constructing a solution network that strictly enforces boundary conditions and a boundary-oriented distance function network, this method ensures applicability to diverse fluid dynamics problems with geometrically complex boundaries. To demonstrate the effectiveness of the proposed method, we conducted tests on two-dimensional incompressible flow fields under different boundary conditions, including steady-state and transient cases.

The main contributions of this work are as follows:

1. We propose a new PINN approach named HB-PINN that are embedded the boundary conditions to solve flow fields in regions with irregular obstructing structures.

2. A power function is proposed to refine the distance metric for handling complex boundary conditions, thereby improving prediction accuracy.

3. Extensive experimental validations on NSE with complex boundary conditions—such as the 2D cylinder wake problem and the 2D obstructed cavity flow with a segmented inlet—demonstrate the superior accuracy of the proposed model compared to existing PINN methods, thereby establishing new benchmarks.

## 2   Related Works

**PINN methods for PDE.** PINN were initially proposed by Raissi et al.[4]. The methods have been extended to diverse domains, including: fluid mechanics[10, 11], medical applications[12, 13], heat transfer[14, 15], and materials science[16, 17]. Furthermore, multiple PINN variants have been developed based on PINN, such as: Modified Fourier Network PINN (MFN-PINN)[18], hp-VPINN[19], Conservative PINN (CPINN)[20]. These variants enhance the generalizability and accuracy of PINN in solving heterogeneous PDE systems.

**Loss conflict problems.** The PINN framework explicitly embeds boundary and initial conditions into the loss function, and the resultant loss conflict problem has garnered significant attention from the research community. A pivotal development by Lu et al. (2021) [21] proposes a distance-informed ansatz that provably satisfies boundary conditions through analytically constructed solutions. This architecture enables exclusive optimization of PDE residuals during training. SA-PINN [22] proposes a spatially adaptive weight matrix to dynamically mitigate gradient conflicts between region-specific loss components during optimization. XPINN [23], as a generalized extension of PINN, introduces an interface-informed decomposition framework for solving nonlinear PDE on arbitrarily complex geometries. Although these methods show significant improvements in loss balancing compared to baseline PINN models, they still suffer from inaccuracies when handling problems with highly complex boundary conditions.

**Complex boundary problems.** Botarelli et al. [24] employed a Modified Fourier Network (MFN-PINN) to solve flow field problems with complex boundaries, demonstrating its superiority through comparisons with Multi-Layer Perceptron-based PINN (MLP-PINN). Sukumar et al. [25] proposed an R-function-based distance function construction method for enforcing boundary conditions in PINN, which significantly improves convenience in first-order derivative-dominated problems. However, the complex boundary analytical distance functions (ADF) constructed by R-functions are not natural functions.

To address the loss term conflicts induced by complex boundary conditions, this work decouples the boundary constraints into two sub-functions: a particular solution function that strictly enforces boundary conditions while being weakly trained on the governing equations, and a distance function describing the spatial proximity to boundaries. Furthermore, to ensure global differentiability of the distance function, we formulate it via a DNN architecture.

## 3 Methodology

### 3.1 Incompressible Navier-Stokes Equations

The incompressible Navier-Stokes equations, excluding external body forces, serve as the governing equations for all flow cases in this study, expressed as:

$$\nabla \cdot \mathbf{u} = 0, \quad \frac{\partial \mathbf{u}}{\partial t} + (\mathbf{u} \cdot \nabla)\mathbf{u} + \frac{1}{\rho}\nabla p - \nu \nabla^2 \mathbf{u} = 0. \tag{1}$$

In the equations, $\mathbf{u}$, $p$, $\rho$, and $\nu$ denote the velocity vector, fluid pressure, density and dynamic viscosity coefficients, respectively. For incompressible flows, $\rho$ and $\nu$ are characteristic fluid parameters that remain spatially and temporally invariant.

### 3.2 HB-PINN Approach

The architecture of our proposed method is illustrated in Fig.1. The framework consists of three subnetworks: $\mathcal{N}_\mathcal{P}$, $\mathcal{N}_\mathcal{D}$, and $\mathcal{N}_\mathcal{H}$. The $\mathcal{N}_\mathcal{P}$ subnetwork (Particular Solution Network) is trained to satisfy boundary conditions, while the $\mathcal{N}_\mathcal{D}$ subnetwork (Distance Metric Network) learns boundary-condition-aware weights by encoding the distance from interior points to domain boundaries. The $\mathcal{N}_\mathcal{H}$ subnetwork (Primary Network) is dedicated to resolving the governing PDE. During the final training phase, only the parameters of $\mathcal{N}_\mathcal{H}$ are optimized, whereas $\mathcal{N}_\mathcal{P}$ and $\mathcal{N}_\mathcal{D}$ remain fixed as pre-trained components. This design effectively addresses challenges involving hybrid interior-exterior boundary configurations. Therefore, we term this methodology Hybrid Boundary Physics-Informed Neural Networks.

In our model, the constraints on the variables are formulated as follows:

$$q(\mathbf{x}, t) = \mathcal{P}_q(\mathbf{x}, t) + \mathcal{D}_q(\mathbf{x}, t) \cdot \mathcal{H}_q(\mathbf{x}, t), \tag{2}$$

here, $q(\cdot)$ denote the physical quantities of interest (e.g., $u$, $v$, and $p$ in a 2D flow model); $\mathcal{P}_q$ represents an additional solution function that satisfies the boundary conditions; $\mathcal{D}_q$ is the distance function; $\mathcal{H}_q$ is the output of the primary network. The distance function $\mathcal{D}_q$ takes a value of 0 at the domain boundaries and rapidly increases to 1 as it moves away from the boundaries. Functionally, $\mathcal{D}_q$ acts as a weight for $\mathcal{H}_q$, modulating its influence across the computational domain.

Under this formulation, the distance function $\mathcal{D}_q$ enforces strict boundary condition compliance by evaluating to zero at domain boundaries while smoothly transitioning to $\mathcal{H}_q$, which satisfies the governing equations in interior regions. When boundary conditions and geometric configurations are simple, both $\mathcal{P}_q$ and $\mathcal{D}_q$ can be analytically expressed. However, analytical expressions for $\mathcal{P}_q$ and $\mathcal{D}_q$ are generally infeasible for complex geometries. Consequently, we leverage three dedicated deep neural networks (DNN) , the previously defined $\mathcal{N}_\mathcal{P}$, $\mathcal{N}_\mathcal{D}$, and $\mathcal{N}_\mathcal{H}$ , to parameterize these components. The composite solution for 2D incompressible flows is thus constructed as:

$$\mathcal{N}_q(\boldsymbol{x}, t) = \mathcal{N}_{\mathcal{P}_q}(\boldsymbol{x}, t) + \mathcal{N}_{\mathcal{D}_q}(\boldsymbol{x}, t) \cdot \mathcal{N}_{\mathcal{H}_u}(\boldsymbol{x}, t). \tag{3}$$

**Sub-network $\mathcal{N}_\mathcal{P}$**

The subnetwork $\mathcal{N}_\mathcal{P}$ adopts the same architecture as conventional PINN, with outputs comprising the velocity components $u$, $v$, and pressure $p$. The loss function for training $\mathcal{N}_\mathcal{P}$ is formulated by integrating residuals of the NSE, boundary conditions (BC), and initial conditions (IC), expressed as:

$$\mathcal{L} = \lambda_1 \mathcal{L}_{\text{PDE}} + \lambda_2 \mathcal{L}_{\text{IC}} + \lambda_3 \mathcal{L}_{\text{BC}}, \tag{4}$$

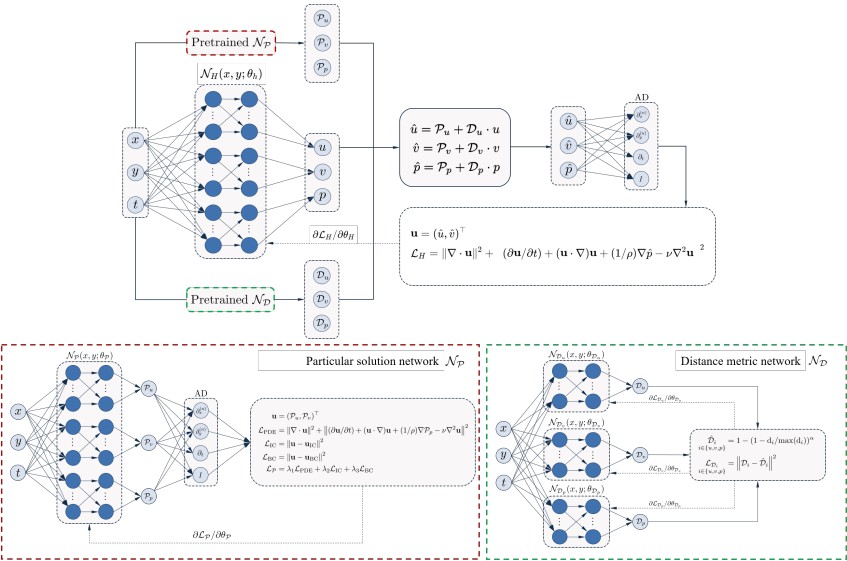

Figure 1: The architecture of the proposed method.

where

$$\mathcal{L}_{\text{BC}} = \frac{1}{N_{\text{BC}}} \sum_{i=1}^{N_{\text{BC}}} \|\mathbf{u}(x_i, t_i) - \mathbf{u}_{\text{BC}}(x_i, t_i)\|^2,  \tag{5}$$

$$\mathcal{L}_{\text{IC}} = \frac{1}{N_{\text{IC}}} \sum_{i=1}^{N_{\text{IC}}} \|\mathbf{u}(x_i, t_0) - \mathbf{u}_{\text{IC}}(x_i, t_0)\|^2,  \tag{6}$$

and

$$\mathcal{L}_{\text{PDE}} = \frac{1}{N_{\text{PDE}}} \sum_{i=1}^{N_{\text{PDE}}} \left( \|\nabla \cdot \mathbf{u}\|^2 + \left\| \frac{\partial \mathbf{u}}{\partial t} + (\mathbf{u} \cdot \nabla) \, \mathbf{u} + \frac{1}{\rho} \nabla p - \nu \nabla^2 \mathbf{u} \right\|^2 \right).  \tag{7}$$

Here, $\lambda_i (i = 1, 2, 3)$ denote the loss weighting coefficients corresponding to each term. Adjusting these weights allows biasing the network's training dynamics to enhance solution accuracy. Our $\mathcal{N}_{\mathcal{P}}$ subnetwork prioritizes boundary and initial condition enforcement by assigning $\lambda_2, \lambda_3 \gg \lambda_1$ (e.g., $\lambda_1 = 1$; $\lambda_2, \lambda_3 = 1000$), enabling preliminary PDE training while strictly satisfying boundary constraints.

By decoupling the training objectives, the subnetwork $\mathcal{N}_{\mathcal{P}}$ focuses exclusively on enforcing boundary conditions through loss weight modulation, without imposing strict constraints on governing equation residuals. This targeted approach circumvents the multi-loss balancing challenges inherent to conventional PINN, while delegating full PDE resolution to the primary network $\mathcal{N}_{\mathcal{H}}$.

## Sub-network $\mathcal{N}_{\mathcal{D}}$

The $\mathcal{N}_{\mathcal{D}}$ subnetwork is a shallow DNN designed to construct distance functions. For each quantity $q$ of interest, we introduce a distance metric network $\mathcal{N}_{\mathcal{D}_q}$. To train this network, sampling points $(x, t)$ are firstly collected from the space-time domain $\Omega \times [0, T]$. The network is then supervised to approximate the signed distance function $\hat{\mathcal{D}}_q(x, t)$ between these points and the boundary associated with $q$, formulated as:

$$\hat{\mathcal{D}}_q = \min(\text{distance to the spatiotemporal boundary of } q).  \tag{8}$$

In order to ensure the network outputs approach 0 near domain boundaries and rapidly increase to 1 in regions far from boundaries, while maintaining dominance of the primary network $\mathcal{N}_{\mathcal{H}_q}$'s predictions in interior regions, we employ a power-law function $f(\hat{\mathcal{D}}_q)$ derived from the distance metric $\hat{\mathcal{D}}_q$ as the training labels for the subnetwork $\mathcal{N}_{\mathcal{D}_q}$. The power-law function $f(\hat{\mathcal{D}}_q)$ is defined as follows:

$$f(\hat{D}_q) = 1 - (1 - \hat{D}_q / \max(\hat{D}_q))^\alpha. \tag{9}$$

The parameter $\alpha$ is a positive value controlling the growth rate of the function. The curves exhibit steeper gradients near boundaries and smoother transitions in regions distant from boundaries. Larger $\alpha$ values enhance the sensitivity of the distance metric network to boundary-proximal points while diminishing its response to interior points, thereby reducing computational demands. However, excessively large $\alpha$ values may lead to erroneous predictions of near-boundary points as having distance values close to 1, which undermines the influence of the specialized solution network in critical regions. To accommodate varying boundary complexities across models, $\alpha$ can be tuned to maintain the steepness of the power-law function within a reasonable range. The loss function for the distance metric network is defined as:

$$\mathcal{L}_{\mathcal{D}_q} = \frac{1}{N_{\mathcal{D}_q}} \sum_{i=1}^{N_{\mathcal{D}_q}} \| \mathcal{D}_q - f\left(\hat{\mathcal{D}}_q\right) \|^2 . \tag{10}$$

Here, $\mathcal{D}_q$ represents the network's output, supervised by the training labels $f(\hat{\mathcal{D}}_q)$. This approach ensures robust training of $\mathcal{N}_{\mathcal{D}_q}$ even under highly complex boundary conditions, while simultaneously enhancing the accuracy of the primary network's training outcomes.

**Sub-network $\mathcal{N}_{\mathcal{H}}$**

The $\mathcal{N}_{\mathcal{H}}$ subnetwork also employs a DNN architecture but with a larger scale compared to the $\mathcal{N}_{\mathcal{P}}$ and $\mathcal{N}_{\mathcal{D}}$ subnetworks. Through pre-training of the $\mathcal{N}_{\mathcal{P}}$ and $\mathcal{N}_{\mathcal{D}}$ subnetworks, boundary conditions are strictly enforced, allowing $\mathcal{N}_{\mathcal{H}}$ to focus solely on minimizing the governing equation residuals. For the 2D incompressible NSE, the loss function of the $\mathcal{N}_{\mathcal{H}}$ subnetwork is defined as:

$$\mathcal{L}_{\mathcal{H}} = \frac{1}{N_{\text{PDE}}} \sum_{i=1}^{N_{\text{PDE}}} \left( \|\nabla \cdot \hat{\mathbf{u}}\|^2 + \left\| \frac{\partial \hat{\mathbf{u}}}{\partial t} + \hat{\mathbf{u}} \cdot \nabla \hat{\mathbf{u}} + \frac{1}{\rho} \nabla \hat{p} - \nu \nabla^2 \hat{\mathbf{u}} \right\|^2 \right) . \tag{11}$$

While this formulation shares the same structure as Eq.7, the variables $\hat{\mathbf{u}}$ and $\hat{p}$ are derived from the modified outputs of Eq.2. This training strategy enables the $\mathcal{N}_{\mathcal{H}}$ subnetwork to focus exclusively on PDE compliance during optimization, effectively eliminating the conflicting gradients between equation residuals and boundary condition losses that commonly plague conventional PINN.

## 4  Experiments

To evaluate the effectiveness of the proposed approach, three two-dimensional incompressible flow cases are investigated, including two steady-state cases and one transient case, to demonstrate the general applicability of our method across different flow fields.

The steady-state and transient incompressible Navier-Stokes equations were solved via the finite element method (FEM), with mesh density ensuring solution accuracy and numerical stability. A time step size of $\Delta t = 0.01$ was adopted for transient simulations. The CFD results obtained under these configurations were utilized as ground truth (GT), the proposed HB-PINN is benchmarked against the conventional soft-constrained PINN (sPINN) [4], hard-constrained PINN (hPINN) [21], modified Fourier Network PINN (MFN-PINN) [18, 24], extended PINN (XPINN) [23], self-adaptive PINN (SA-PINN) [22], and PirateNets [26]. Notably, the hPINN, proposed by Lu et al.[21], exhibits erratic behavior in scenarios with complex boundary conditions, often failing to produce reliable results for comparative analysis. To ensure comparability of results, the comparative analysis of the hPINN method in this study selectively relaxes the enforcement of boundary conditions in specific subdomains, as exemplified by the near-inlet wall regions in Case 2 ($0 < x < 0.4$, $y = 0$ & $y = 1$). The detailed rationale for relaxing boundary conditions is provided in Appendix D. Additionally, to

assess the impact of our HB-PINN, we conducted three ablation studies on Case 2, the results are provided in Appendix C.

In this section, the experimental setup is introduced first, followed by three case studies: steady-state two-dimensional flow around a cylinder, steady-state flow in a segmented inlet with an obstructed square cavity, and transient flow in a segmented inlet with an obstructed square cavity.

## 4.1 Experimental Settings

As illustrated in Fig. 1, the proposed HB-PINN takes space-time coordinates as inputs and outputs velocity components $(u, v)$ and pressure $p$. In the implementation phase, the training process employs a distributed architecture: each subnetwork ($\mathcal{N}_{\mathcal{P}}$, $\mathcal{N}_{\mathcal{D}}$, and $\mathcal{N}_{\mathcal{H}}$) utilizes three independent DNN dedicated to variables $u$, $v$, and $p$, resulting in a total of nine distinct DNN. After obtaining separate predictions for $u$, $v$, and $p$, these outputs are integrated through Eq. 3 to reconstruct the composite solution. For detailed training protocols refer to Appendix B.

In the models discussed in this study, the NSE are non-dimensionalized using characteristic scales. For the three benchmark cases analyzed in the main text, the Reynolds number is uniformly set to $Re = 100$ as the baseline configuration. Comparative studies at higher Reynolds numbers ($Re = 500, 1000, 2000$) are systematically documented in Appendix E.

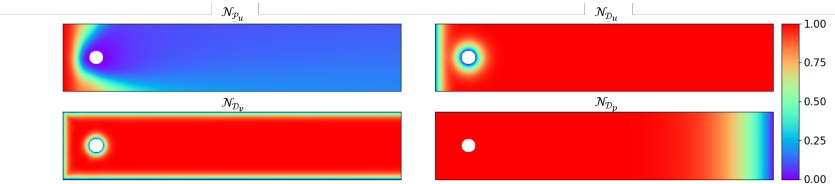

Figure 2: The trained boundary prediction for the Case 1. $\mathcal{N}_{\mathcal{P}_u}$ represents the result of the particular solution network for $u$, while $\mathcal{N}_{\mathcal{D}_u}$, $\mathcal{N}_{\mathcal{D}_v}$, and $\mathcal{N}_{\mathcal{D}_p}$ respectively represent the results of the distance metric network for $u$, $v$, and $p$.

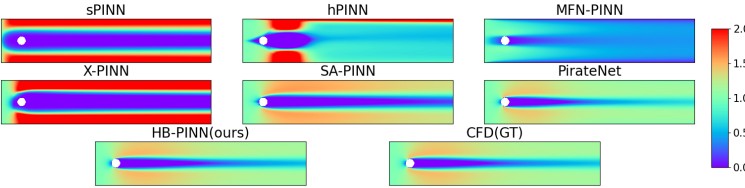

Figure 3: Velocity distributions for the flow around a cylinder from different methods.

## 4.2 Case 1: Two-dimensional flow around a cylinder

The flow past a circular cylinder represents a canonical benchmark in fluid mechanics(see Appendix A.1 for detailed geometric specifications). The boundary conditions are specified as follows:

$$u = 1, \quad x = 0, \ 0 \le y \le 1; \quad u \cdot n_x = 0, \quad \text{others};$$
$$v \cdot n_y = 0, \quad \text{on } \partial\Omega; \quad p = 0, \quad x = 5, \ 0 \le y \le 1.$$

Following Eq.4, the loss function for the $\mathcal{N}_{\mathcal{P}}$ subnetwork in the proposed model is formulated as:

$$\mathcal{L}_{\mathcal{P}} = \lambda_1 \Big( \|\nabla \cdot \hat{u}\|^2 + \|\partial\hat{u}/\partial t + (\hat{u} \cdot \nabla)\,\hat{u} + 1/\rho \nabla \hat{p} -$$
$$\nu \nabla^2 \hat{u}\big\|^2_{x \in \Omega} \Big) + \lambda_2 \left( \|\mathcal{U}_{\mathcal{P}} - \mathcal{U}_B\|^2_{x \in \partial\Omega} + \|p_{\mathcal{P}} - p_B\|^2_{x \in \partial\Omega} \right).$$

Here, $\Omega$ denotes the interior of the geometric region, while $\partial\Omega$ denotes the boundary. The loss weight $\lambda_1$ for the equation part is set to 1, and the loss weight $\lambda_2$ for the boundary part is set to 1000, with the aim of making the network focus more on the training of boundary conditions.

The outputs of the $\mathcal{N}_\mathcal{P}$ and $\mathcal{N}_\mathcal{D}$ networks are shown in Fig.2. The final modification to the variable result is as follows:

$$\hat{u} = \mathcal{N}_{\mathcal{P}_u}(x,t) + \mathcal{N}_{\mathcal{D}_u}(x,t) \cdot \mathcal{N}_{\mathcal{H}_u}(x,t), \hat{v} = \mathcal{N}_{\mathcal{D}_v}(x,t) \cdot \mathcal{N}_{\mathcal{H}_v}(x,t), \text{ and } \hat{p} = \mathcal{N}_{\mathcal{D}_p}(x,t) \cdot N_{\mathcal{H}_p}(x,t).$$

Since both $v$ and $p$ are set to 0 at the boundary, there is no need to configure the boundary results separately; they are simply constrained by the distance function.

In the case of 2D cylindrical pipe flow model, the qualitative results from sPINN, hPINN, MFN-PINN, XPINN, SA-PINN, PirateNet, HB-PINN are shown in Fig.3. The residuals of these methods compared with CFD results are shown in Fig.4. More detailed error metrics are summarized in the "Case1" section of Table 1.

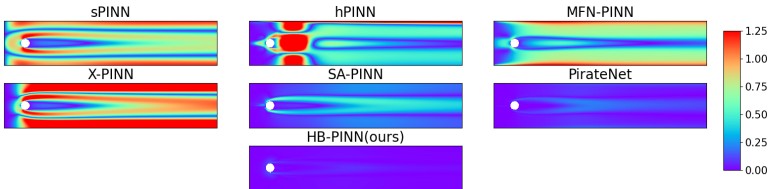

Figure 4: The residuals of sPINN, hPINN, MFN-PINN, XPINN, SA-PINN, PirateNet, and our HB-PINN compared to the GT in Case 1.

### 4.3 Case 2: steady-state flow in a segmented inlet with an obstructed square cavity

The model incorporates two rectangular obstructions positioned along the top and bottom walls of a square cavity, with a segmented inlet configuration (see Appendix A.2 for detailed geometric specifications). The boundary conditions are configured as follows:

$$u = 0.5, \quad x = 0, \ y \in [0, 0.2] \cup [0.4, 0.6] \cup [0.8, 1]; \quad u = 0, \text{ others;}$$
$$v = 0, \text{ on } \partial\Omega; \quad p = 0, \ x = 1, \ y \in [0.8, 1].$$

The output results of the boundary condition solution function network related to $u$ and the distance function network for $u$, $v$, and $p$ are shown in Fig.5.

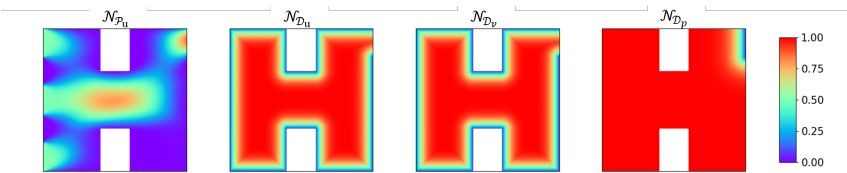

Figure 5: The trained boundary prediction for Case 2. $\mathcal{N}_{P_u}$ represents the result of the particular solution network for $u$, while $\mathcal{N}_{D_u}$, $\mathcal{N}_{D_v}$, and $\mathcal{N}_{D_p}$ respectively represent the results of the distance metric network for $u$, $v$, and $p$.

The velocity distributions for case 2 from sPINN, hPINN, MFN-PINN, SA-PINN, XPINN, PirateNet and our HB-PINN are shown in Fig.6. Compared to existing methods, the HB-PINN framework demonstrates superior capability in simulating models with complex inflow boundary conditions, achieving a MSE reduction of an order of magnitude relative to CFD benchmarks. The residuals of these methods compared with CFD results are shown in Fig. 7. More detailed error metrics are summarized in the "Case2" section of Table 1.

Based on Case 2, we conduct ablation studies on two critical pre-trained sub-networks by evaluating the outcomes of separately employing the specific solution network and the distance metric network. Here, "mP" denotes using only the particular solution network, while "mD" refers to using only the distance metric network.

Under the mP approach, the distance function is normalized solely based on the distance to the boundary without adjustment using power functions, resulting in lower accuracy within the domain. Under the mD approach, while internal errors are reduced during training, the enforcement of

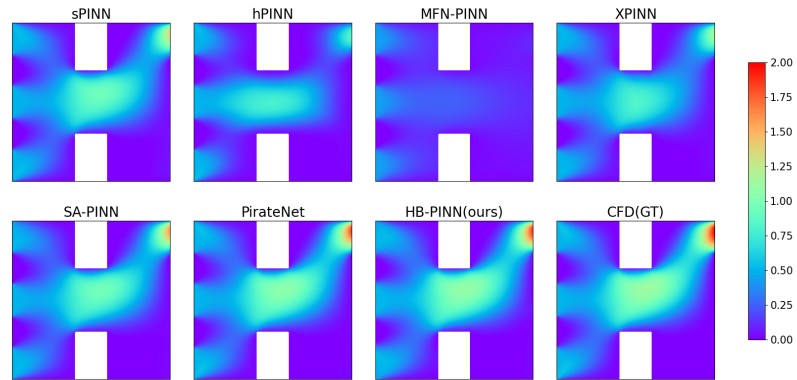

Figure 6: Comparison of velocity distributions for Case 2 across different methods.

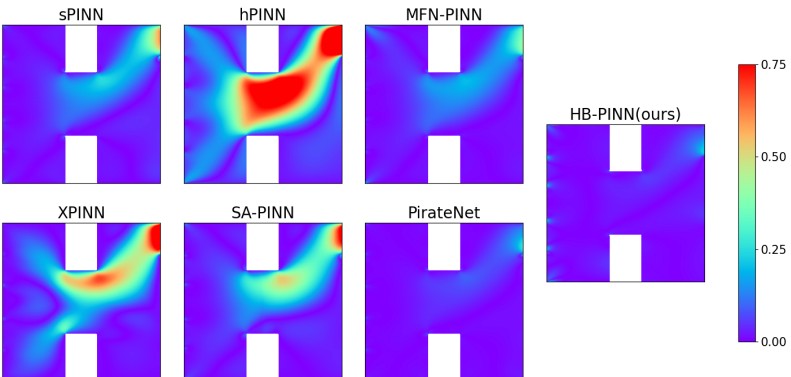

Figure 7: The residuals of sPINN, hPINN, MFN-PINN, XPINN, SA-PINN, PirateNet, and our HB-PINN compared to the GT in Case 2.

boundary conditions is imprecise, which affects the final results. These tests demonstrate that both constructive functions in our method contribute positively to the solution process, further validating the necessity of applying both functions simultaneously. The comparison of residuals is shown in Fig.8. For a more detailed error analysis, please refer to Table2 in Appendix C.

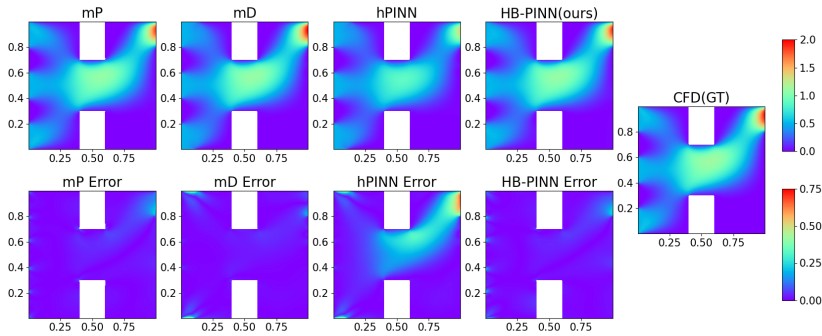

Figure 8: Comparison of velocity results and residuals between the mP and mD methods in Case 2.

## 4.4 Case 3: transient flow in a segmented inlet with an obstructed square cavity

The difference between this model and case 2 lies in the consideration of the transient situation, which further increases the complexity of the boundary conditions. The boundary conditions are set

as follows: the inlet normal velocity $u$ is 0.5, the outlet static pressure is 0, and the velocity at other boundaries is set to 0, satisfying the no-slip condition. The initial conditions are set with the velocity components $u$, $v$ and the pressure $p$ all equal to 0. Relative to Case 2, the addition of initial conditions:

$$u = 0, v = 0, p = 0, \quad t = 0$$

The network structure parameters chosen for the transient case are the same as those for case 2. Fig. 9 shows the velocity results obtained using different methods at $t = 1$. In the transient case, due to the increased complexity of the boundary conditions, the accuracy of the other methods significantly decreases, while HB-PINN maintains a high level of accuracy. The velocity results and the error compared to the CFD method are shown in Fig. 10. These results demonstrate the versatility of HB-PINN in both steady-state and transient situations.

Table 1: Comparative analysis of error metrics across three Cases. Among the baseline methods, PirateNet achieves the highest accuracy, while HB-PINN represents our proposed methodology.

| Case | Method | Error Metrics | | |
| --- | --- | --- | --- | --- |
| | | MSE($\downarrow$) | MAE($\downarrow$) | Relative L2($\downarrow$) |
| Case 1 | sPINN | 0.520 78 | 0.642 16 | 0.668 05 |
| | hPINN | 0.583 47 | 0.527 29 | 0.707 12 |
| | MFN-PINN | 0.423 73 | 0.574 10 | 0.213 63 |
| | XPINN | 1.119 24 | 0.909 27 | 0.979 36 |
| | SA-PINN | 0.053 25 | 0.189 67 | 0.213 63 |
| | PirateNet | 0.007 63 | 0.066 89 | 0.080 89 |
| | HB-PINN | **0.004 33** | **0.020 12** | **0.060 93** |
| Case 2 | sPINN | 0.008 01 | 0.051 73 | 0.200 36 |
| | hPINN | 0.040 60 | 0.112 42 | 0.451 10 |
| | MFN-PINN | 0.116 26 | 0.211 86 | 0.763 37 |
| | XPINN | 0.021 49 | 0.083 85 | 0.328 24 |
| | SA-PINN | 0.005 42 | 0.045 80 | 0.164 95 |
| | PirateNet | 0.001 86 | 0.023 57 | 0.096 79 |
| | HB-PINN | **0.000 88** | **0.018 88** | **0.066 55** |
| Case 3 | sPINN | 0.124 00 | 0.218 76 | 0.788 19 |
| | hPINN | 0.155 59 | 0.239 51 | 0.882 88 |
| | MFN-PINN | 0.105 43 | 0.201 50 | 0.726 78 |
| | XPINN | 0.041 49 | 0.120 28 | 0.455 92 |
| | SA-PINN | 0.106 12 | 0.196 42 | 0.729 15 |
| | PirateNet | 0.032 02 | 0.106 82 | 0.400 54 |
| | HB-PINN | **0.008 25** | **0.048 70** | **0.203 32** |

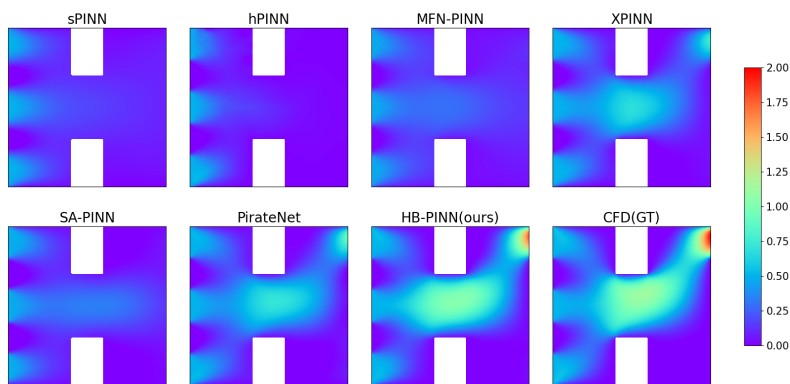

Figure 9: Comparison of velocity distributions at t = 1 for different methods in Case 3.

## 4.5 Result Discussion

The quantitative results of the error metrics between CFD calculation and the predictions from sPINN, hPINN, MFN-PINN, SA-PINN, XPINN, PirateNet and HB-PINN for the three cases are presented in Table 1. Combined with the qualitative results shown in Fig.3,4,6,7,9,10, it is shown that our HB-PINN can accurately solve complex boundary flow problems with interior barriers, while the traditional soft-constrained PINN (sPINN) method and hard-constrained PINN (hPINN) method incur significant errors in solving such problems. The training results of SA-PINN and XPINN show improvements in accuracy; however, significant errors persist in the cases with more complex boundary conditions.

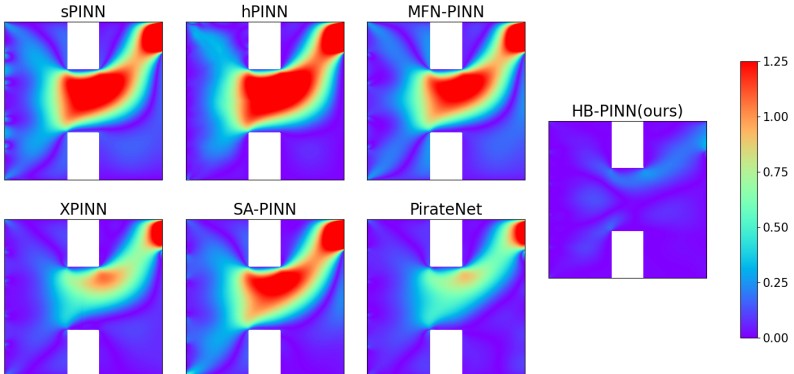

Figure 10: The residuals of sPINN, hPINN, MFN-PINN, XPINN, SA-PINN, PirateNet, and our HB-PINN compared to the GT in Case 3.

## 5   Limitations

The optimal values of critical parameters in the methodology, such as the $\alpha$ parameter in the distance metric network and the training iterations of the particular solution network, are currently determined empirically. Further research is required to systematically identify their optimal configurations.

## 6   Conclusion

PINN hold great potential as surrogate models for fluid dynamics; however, their application is often hindered by the complexity of the NSEs and intricate boundary conditions. This paper introduces a novel method, HB-PINN, which features a primary network focused on inner domain points and a distance metric network that enhances predictions at the boundaries, ensuring accurate solutions for both boundary and interior regions. This makes the approach applicable to be adaptable to more complex fluid dynamics models. Through tests on steady-state two-dimensional flow around a cylinder, steady-state segmented inlet with obstructed square cavity flow, and transient segmented inlet with obstructed square cavity flow, our method outperforms typical PINN approaches by a large margin, establishing itself as a benchmark for solving NSEs with complex boundaries. Additional analysis is provided in the appendices to further illustrate the effectiveness of our method.

## Acknowledgements

The authors would like to acknowledge the financial support received from Donghai Laboratory (2024SSYS0091), Open Fund of Beijing Key Laboratory of Advanced Optical Remote Sensing Technology (AORS202408), National Natural Science Foundation of China (12304346) , Development Program of the Department of Science and Technology of Shaanxi Province (2025CY-YBXM-073), and Haina Plan of Zhouchuang Future from Mei Lin.

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

# A  Supplementary Geometric Details

## A.1  Case1:Flow around a Cylinder

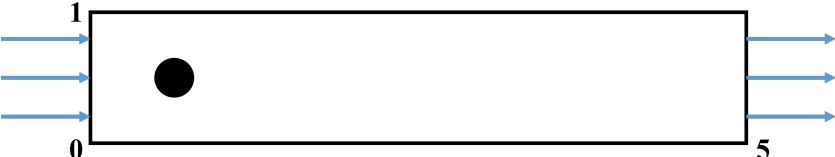

Figure 11: Scheme of the two-dimensional flow around a cylinder.

Case 1 corresponds to the canonical flow past a circular cylinder. As schematically depicted in Fig.11. Following non-dimensionalization, the computational domain is configured with a length of 5 and a width of 1. A circular obstruction with diameter $D = 0.2$ is centered at $(x, y) = (0.5, 0.5)$. The boundary conditions are defined as follows:

- **Inlet (Left Boundary):** Horizontal inflow velocity $u = 1$, $v = 0$.
- **Outlet (Right Boundary):** Static pressure fixed at $p = 0$.
- **Top/Bottom Boundaries:** No-slip walls ($u = v = 0$).

This configuration corresponds to a Reynolds number $Re = 100$, with the dimensionless density $\rho = 1$. The dynamic viscosity $\nu$ is determined by the Reynolds number formula $Re = \frac{uD}{\nu}$, where $u$ denotes the characteristic velocity and $D$ the characteristic length.

## A.2  Case2:Steady-State Flow in a Segmented Inlet with an Obstructed Square Cavity

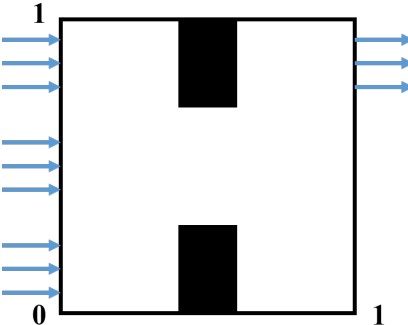

Figure 12: Scheme of the Segmented inlet with obstructed square cavity flow.

The model features two rectangular obstructions positioned at the top and bottom walls of a square cavity with an edge length of 1 unit. As illustrated in Fig.12, the inlet is divided into three segments, and the obstructions have a width of 0.2 units and a height of 0.3 units. The boundary conditions are defined as follows:

- **Inlet:** Normal inflow velocity $u = 0.5$, $v = 0$.
- **Outlet:** Static pressure set to $p = 0$.
- **Other Boundaries:** No-slip conditions ($u = v = 0$) on all walls and obstruction surfaces.

In Case 3 of the main text, the geometric configuration remains identical to that of Case 2, with the sole distinction lying in the incorporation of transient flow dynamics.

# B Training Details

The framework is developed on the PyTorch machine learning platform, and training iterations are accelerated using a high-performance GPU.

## B.1 Case1

**Subnetwork $\mathcal{N}_{\mathcal{P}}$:**

- Network architecture: Three separate deep neural networks (DNNs) for variables $u$, $v$, and $p$, each structured as $[2] + 6 \times [64] + [1]$
- Training epochs: 10,000
- Activation function: tanh
- Optimizer: Adam
- Learning rate: $1 \times 10^{-3}$

**Subnetwork $\mathcal{N}_{\mathcal{D}}$:**

- Network architecture: Three separate DNNs for $u$, $v$, and $p$, each structured as $[2] + 4 \times [20] + [1]$
- Training epochs: 300,000
- Activation function: tanh
- Optimizer: Adam
- Learning rate: Initial learning rate of $1 \times 10^{-3}$ with a learning rate annealing strategy
- $\alpha = 10$

**Subnetwork $\mathcal{N}_{\mathcal{H}}$:**

- Network architecture: Three separate DNNs for $u$, $v$, and $p$, each structured as $[2] + 6 \times [128] + [1]$
- Training epochs: 300,000
- Activation function: tanh
- Optimizer: Adam
- Learning rate: Initial learning rate of $1 \times 10^{-3}$ with a learning rate annealing strategy

## B.2 Case2

**Subnetwork $\mathcal{N}_{\mathcal{P}}$:**

- Network architecture: Three separate deep neural networks (DNNs) for variables $u$, $v$, and $p$, each structured as $[2] + 6 \times [64] + [1]$
- Training epochs: 10,000
- Activation function: tanh
- Optimizer: Adam
- Learning rate: $1 \times 10^{-3}$

**Subnetwork $\mathcal{N}_{\mathcal{D}}$:**

- Network architecture: Three separate DNNs for $u$, $v$, and $p$, each structured as $[2] + 4 \times [64] + [1]$
- Training epochs: 300,000
- Activation function: tanh
- Optimizer: Adam
- Learning rate: Initial learning rate of $1 \times 10^{-3}$ with a learning rate annealing strategy
- $\alpha = 5$

**Subnetwork $\mathcal{N}_{\mathcal{H}}$:**

- Network architecture: Three separate DNNs for $u$, $v$, and $p$, each structured as $[2]+6\times[128]+[1]$
- Training epochs: 500,000
- Activation function: tanh
- Optimizer: Adam
- Learning rate: Initial learning rate of $1\times10^{-3}$ with a learning rate annealing strategy

### B.3 Case3

**Subnetwork $\mathcal{N}_{\mathcal{P}}$:**

- Network architecture: Three separate deep neural networks (DNNs) for variables $u$, $v$, and $p$, each structured as $[3]+6\times[64]+[1]$
- Training epochs: 10,000
- Activation function: tanh
- Optimizer: Adam
- Learning rate: $1\times10^{-3}$

**Subnetwork $\mathcal{N}_{\mathcal{D}}$:**

- Network architecture: Three separate DNNs for $u$, $v$, and $p$, each structured as $[3]+4\times[64]+[1]$
- Training epochs: 300,000
- Activation function: tanh
- Optimizer: Adam
- Learning rate: Initial learning rate of $1\times10^{-3}$ with a learning rate annealing strategy
- $\alpha=5$

**Subnetwork $\mathcal{N}_{\mathcal{H}}$:**

- Network architecture: Three separate DNNs for $u$, $v$, and $p$, each structured as $[3]+6\times[128]+[1]$
- Training epochs: 500,000
- Activation function: tanh
- Optimizer: Adam
- Learning rate: Initial learning rate of $1\times10^{-3}$ with a learning rate annealing strategy

### B.4 Other PINN methods

- Network architecture: Three separate DNNs for $u$, $v$, and $p$, each structured as $[2]+6\times[128]+[1]$
- Training epochs: 500,000
- Activation function: tanh
- Optimizer: Adam
- Learning rate: Initial learning rate of $1\times10^{-3}$ with a learning rate annealing strategy
- Points per epoch: 20,000 (random sampling from domain)

## C  Ablation studies

The three ablation studies are as follows:

1. Comparative analysis of mP (particular solution network only) and mD (distance metric network only) against hPINN and HB-PINN, where mP and mD represent experimental configurations using isolated components of HB-PINN.

2. Impact assessment of varying pre-training epochs for the particular solution network ($\mathcal{N}_\mathcal{P}$) on the final accuracy of the primary network.

3. Sensitivity analysis of the power-law exponent $\alpha$ in the distance metric network ($\mathcal{N}_\mathcal{D}$), which governs the transition steepness of the distance function, to evaluate its influence on solution accuracy.

### C.1 mP and mD

In this section, we evaluate the outcomes of separately employing the particular solution network and the distance metric network, where mP refers to using only the particular solution network and mD refers to using only the distance metric network. The experimental results are compared with hPINN and HB-PINN, and velocity predictions are shown in Fig.13.

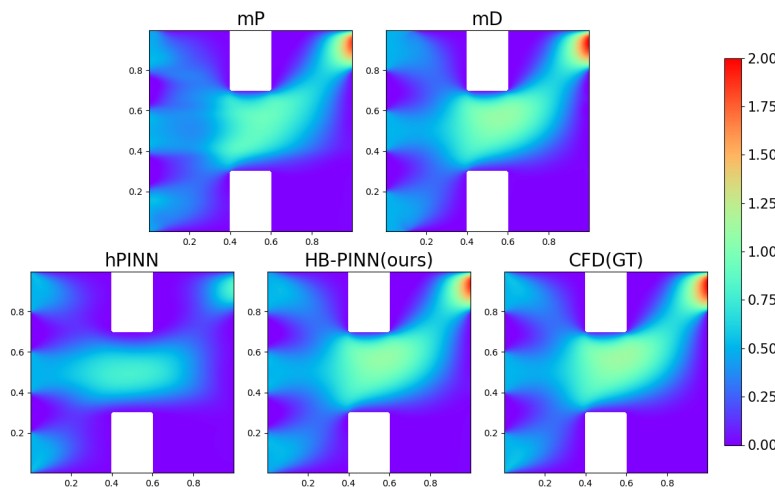

Figure 13: Comparison of velocity results of the mP and mD methods in Case 2.

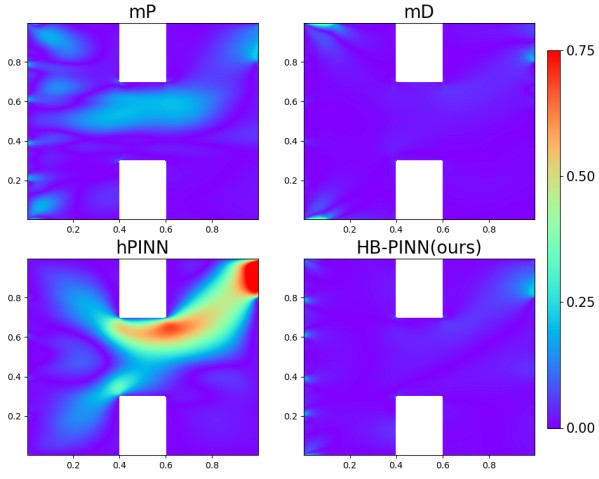

Figure 14: The residuals of mP, mD, hPINN,and HB-PINN compared to the GT in Case 2.

Under the mP approach, the distance function is normalized solely based on the distance to the boundary without adjustment using power functions, resulting in lower accuracy within the domain.

Under the mD approach, while internal errors are reduced during training, the enforcement of boundary conditions is imprecise, which affects the final results. These tests demonstrate that both constructive functions in our method contribute positively to the solution process, further validating the necessity of applying both functions simultaneously. The comparison of residuals is shown in Fig.14.More detailed error metrics are summarized in the "mP and mD" section of Table **??**.

### C.2   Particular solution network under different training epochs

In the HB-PINN method, the particular solution network ($\mathcal{N}_\mathcal{P}$) is trained with a shallow deep neural network (DNN) using a soft constraint approach, aiming to provide a pre-trained solution that satisfies the boundary conditions, by tuning the weights associated with the loss terms, the network's output is rigorously constrained to strictly satisfy the boundary conditions. When the weight of the boundary condition loss term in the loss function is set to an extremely high value (e.g., 1000), excessive training iterations may lead to overfitting in local boundary regions, thereby compromising the global smoothness of the network's output. To mitigate this issue, a strategy with limited training iterations is generally adopted. In this section, we supplement our analysis by investigating the impact of different training iterations (10,000, 30,000, and 50,000) of $\mathcal{N}_\mathcal{P}$ on the final training results. The velocity results and their residuals relative to the CFD method are shown in Fig.15,16.

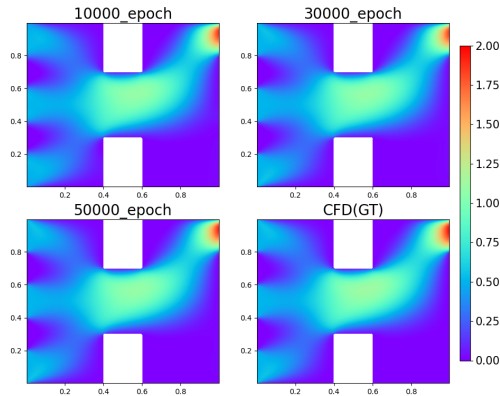

Figure 15: Final velocity results for $\mathcal{N}_\mathcal{P}$ trained with 10,000, 30,000, and 50,000 iterations.

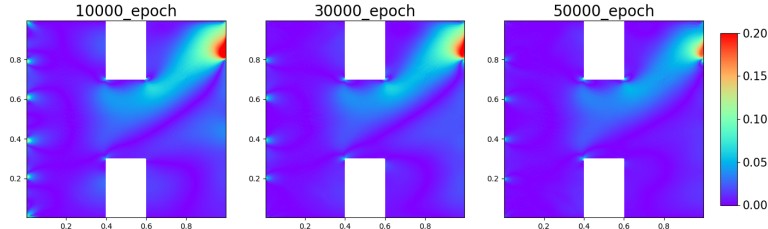

Figure 16: Residuals of the final velocity results compared to the CFD method for $\mathcal{N}_\mathcal{P}$ trained with 10,000, 30,000, and 50,000 iterations.

As evidenced by the error metrics in Figure 15 and the "Epoch for $\mathcal{N}_\mathcal{P}$" section of Table 2, limiting the training iterations of the particular solution subnetwork ($\mathcal{N}_\mathcal{P}$) leads to lower global error within the computational domain and more rapid convergence of the primary network ($\mathcal{N}_\mathcal{H}$). However, this configuration results in reduced precision in boundary regions compared to cases where $\mathcal{N}_\mathcal{P}$ undergoes extended training.Currently, the number of training iterations for $\mathcal{N}_\mathcal{P}$ is determined empirically, and further research is needed to balance accuracy and convergence speed optimally.

## C.3 Distance metric network under different $\alpha$

This section evaluates the impact of four critical parameter values ($\alpha = 3, 5, 10, 15$) in the distance metric network on the predictive performance of the HB-PINN method. Notably, $\alpha = 5$ corresponds to the parameter configuration adopted in Case 2 of the main text. The velocity prediction results, as illustrated in Fig.17, demonstrate the sensitivity of model accuracy to variations in $\alpha$.

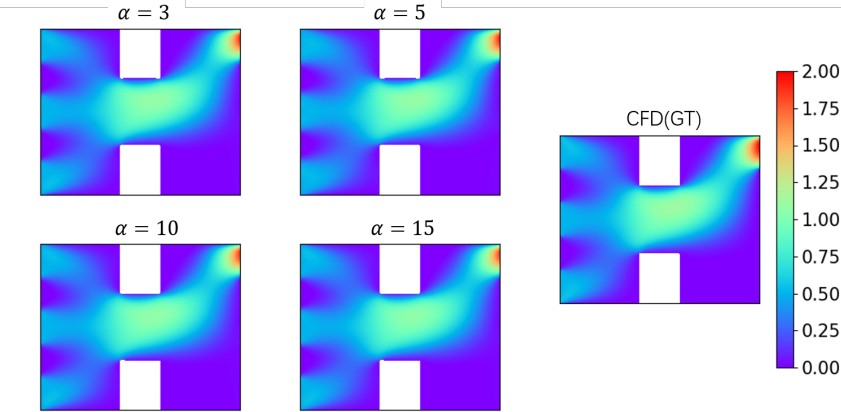

Figure 17: Comparison of velocity results between HB-PINN and CFD for $\alpha = 3, 5, 10,$ and $15$.

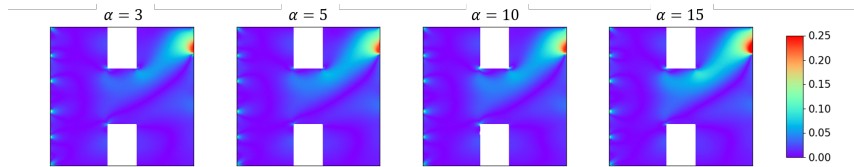

Figure 18: Residuals between HB-PINN and CFD velocity results for $\alpha = 3, 5, 10,$ and $15$.

The parameter $\alpha$ governs the steepness of the transition in the distance function near boundaries. Excessively small $\alpha$ values (e.g., $\alpha = 3$) result in insufficient sensitivity to boundary constraints, while overly large $\alpha$ values (e.g., $\alpha = 10, 15$) lead to excessively thin boundary layers, thereby amplifying training challenges in boundary regions. The velocity prediction residuals compared to CFD results and the statistical metrics of the mean squared error (MSE) are shown in Fig.18 and the $\alpha$ for $\mathcal{N}_\mathcal{D}$ section of Table2. These results indicate that both undersized and oversized $\alpha$ values degrade prediction accuracy. However, the selection of $\alpha$ currently remains empirically determined, and identifying an optimal $\alpha$ value warrants further investigation.

## C.4 Error Records

# D Supplementary Details on Boundary Condition Relaxation in the hPINN Method

In the hPINN method, using only boundary conditions to constrain the solution network $\mathcal{N}_\mathcal{P}$ is effective when the boundary conditions are simple. However, for complex boundaries, this approach leads to distorted or discontinuous outputs within the solution domain, especially near junctions between different boundary types.

Fig.19 presents the results of the boundary particular solution network $\mathcal{N}_\mathcal{P}$ from the original hPINN, the hPINN with partially relaxed boundary constraints, and the HB-PINN. It can be clearly observed that under the original hPINN framework, the output of the $\mathcal{N}_\mathcal{P}$ network exhibits irregular solutions within the domain. Since it is trained solely on boundary conditions, its internal results are uncontrollable. This significantly increases the training difficulty of the main network, even hindering

Table 2: Comparative analysis of error metrics across ablation studies

| Category | Configuration | Error Metrics($\downarrow$) | | |
|---|---|---|---|---|
| | | MSE($\downarrow$) | MAE($\downarrow$) | Relative L2($\downarrow$) |
| mP & mD | hPINN | 0.040 60 | 0.112 42 | 0.451 10 |
| | $mP$ | 0.004 84 | 0.045 89 | 0.155 75 |
| | $mD$ | 0.004 09 | 0.022 69 | 0.143 29 |
| | HB-PINN | 0.000 88 | 0.018 88 | 0.066 55 |
| Epoch for $\mathcal{N}_{\mathcal{P}}$ | 10000 | 0.000 88 | 0.018 88 | 0.066 55 |
| | 30000 | 0.002 34 | 0.019 07 | 0.108 33 |
| | 50000 | 0.002 48 | 0.016 44 | 0.111 69 |
| $\alpha$ for $\mathcal{N}_{\mathcal{D}}$ | 3 | 0.004 71 | 0.025 77 | 0.153 74 |
| | 5 | 0.000 88 | 0.018 88 | 0.066 55 |
| | 10 | 0.001 77 | 0.023 12 | 0.094 29 |
| | 15 | 0.001 40 | 0.021 99 | 0.084 03 |

convergence – as we observed in our hPINN experiments: strict constraints on certain boundary regions had to be relaxed for the main network loss to decrease effectively. The loss records in Table3 fully demonstrate this point. In Relaxed hPINN, we relaxed the boundary constraints in the regions $0 < x < 0.4$, $y = 0$ & $y = 1$, which indeed effectively promoted the reduction of loss during main network training. However, this compromise violates the requirement for full boundary condition satisfaction, which is unacceptable for solving physical models. This precisely highlights the dilemma of traditional hard-constraint methods when handling complex boundary problems. Therefore, we introduced the $\mathcal{N}_{\mathcal{P}}$ with weak equation constraints as an alternative solution aimed at more robustly handling complex boundary constraints.

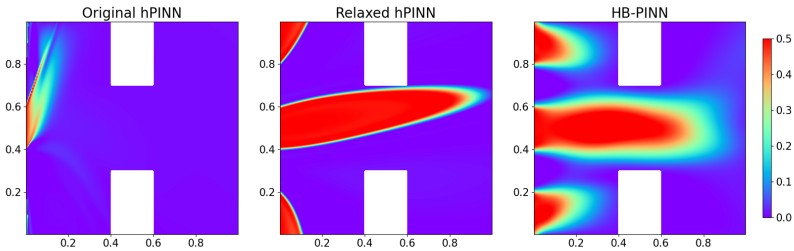

Figure 19: Comparative Analysis of Particular Solution Network Results across Original hPINN, Relaxed hPINN, and HB-PINN

Table 3: Loss comparison across different methods at various epochs

| Epoch | Original hPINN | Relaxed hPINN | HB-PINN |
|---|---|---|---|
| 0 | 2 335 810 | 47.44 | 7.129 |
| 1000 | 15 292 | 29.82 | 2.517 |
| 2000 | 1 855 618 | 30.75 | 1.797 |
| 3000 | 5 720 168 | 28.45 | 1.588 |

# E  Supplementary details of the transient case (Case 3).

In the main text of Case 3, we present the velocity results at t = 1 for the transient segmented inlet with obstructed square cavity flow model. At t = 1, the flow can be considered to have developed to a state approaching steady-state results. Fig.20,21,22,23 display the velocity results at several time

steps during the development of the transient model, specifically at t = 0.1, t = 0.3, t = 0.5, and t = 0.7, to demonstrate that our HB-PINN method achieves high accuracy across all time steps when solving the transient model. The errors in velocity relative to the CFD method are shown in Fig.24,25,26,27

The velocity results at different time steps and their corresponding mean square errors relative to the CFD method are quantitatively summarized in Table4.

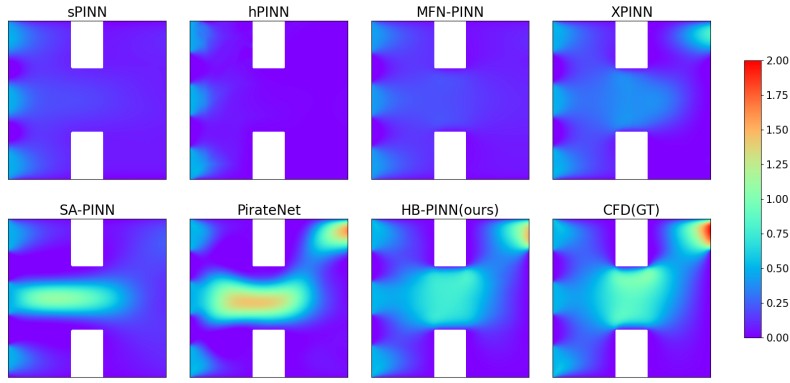

Figure 20: Comparison of velocity distributions at t = 0.1 for different methods in Case 3.

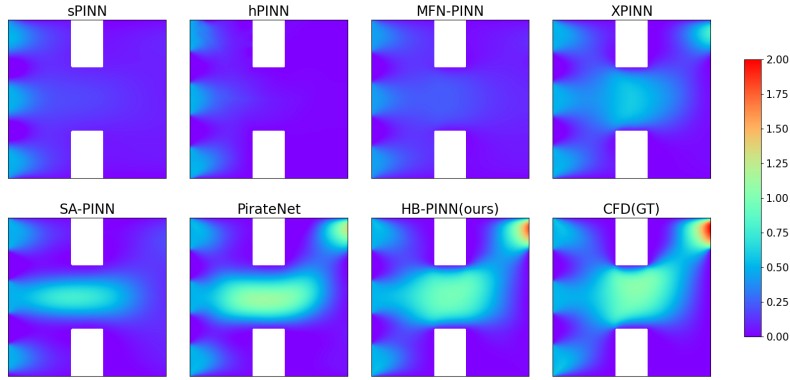

Figure 21: Comparison of velocity distributions at t = 0.3 for different methods in Case 3.

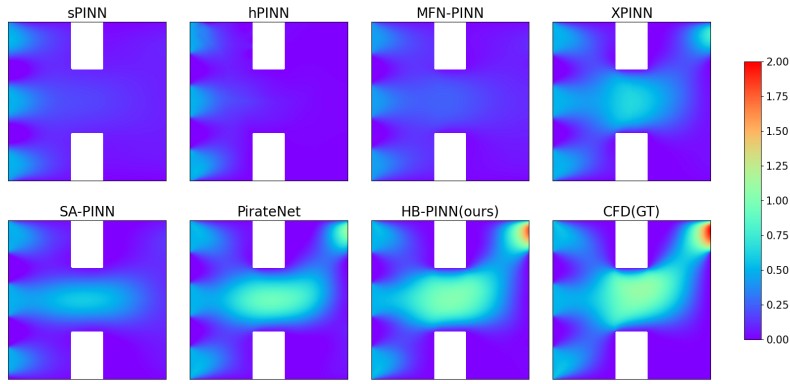

Figure 22: Comparison of velocity distributions at t = 0.5 for different methods in Case 3.

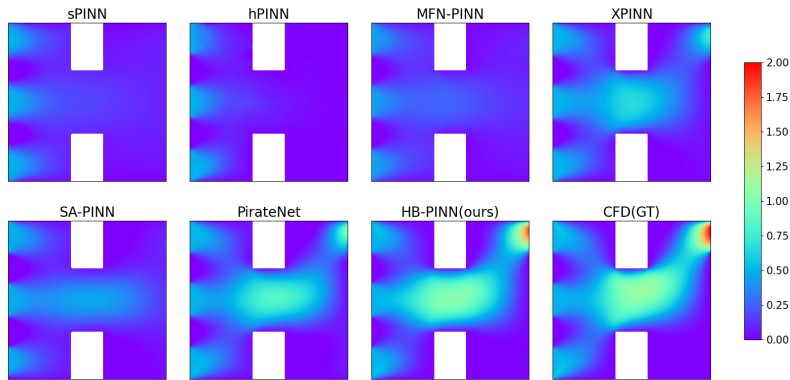

Figure 23: Comparison of velocity distributions at t = 0.7 for different methods in Case 3.

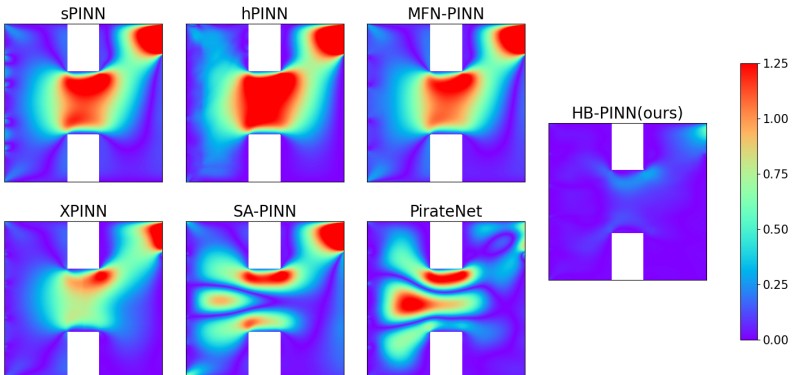

Figure 24: Comparison of the residuals relative to the ground truth (GT) for sPINN, hPINN, SA-PINN, XPINN, and our HB-PINN at t = 0.1 in Case 3.

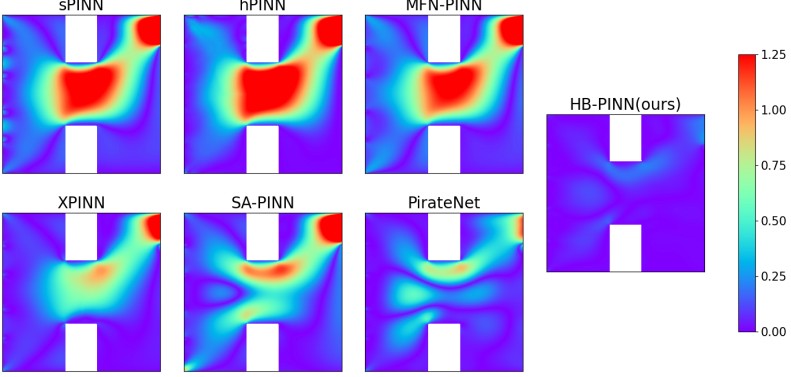

Figure 25: Comparison of the residuals relative to the ground truth (GT) for sPINN, hPINN, SA-PINN, XPINN, and our HB-PINN at t = 0.3 in Case 3.

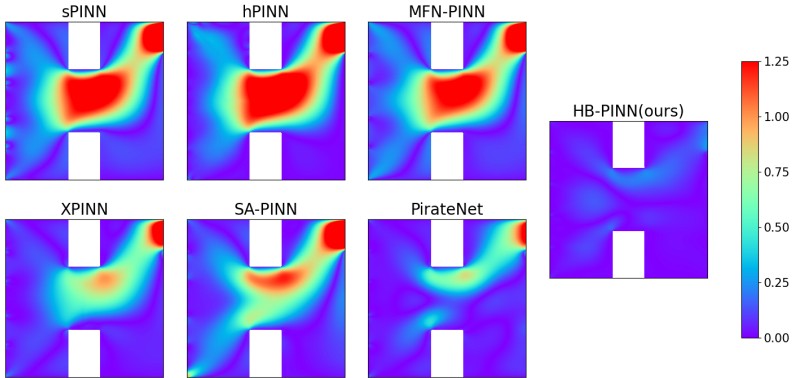

Figure 26: Comparison of the residuals relative to the ground truth (GT) for sPINN, hPINN, SA-PINN, XPINN, and our HB-PINN at t = 0.5 in Case 3.

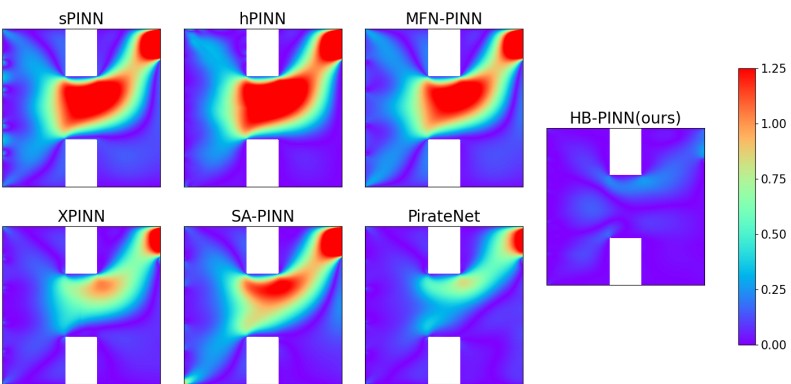

Figure 27: Comparison of the residuals relative to the ground truth (GT) for sPINN, hPINN, SA-PINN, XPINN, and our HB-PINN at t = 0.7 in Case 3.

## F Cross-sectional profiles and time-history results in high-gradient regions

This section compares the velocity and pressure predictions of sPINN, PirateNet, and HB-PINN in high-gradient regions. Fig.28 displays the predictive results along the cross-section at $y = 0.5$ in Case 2 compared with CFD results, while Fig.29 presents the temporal distribution of predictions at the center point $(0.5, 0.5)$ in Case 3 against CFD references. Both regions represent areas with substantial gradients within their respective domains. The results demonstrate that HB-PINN achieves optimal prediction accuracy for all physical quantities in both test cases.

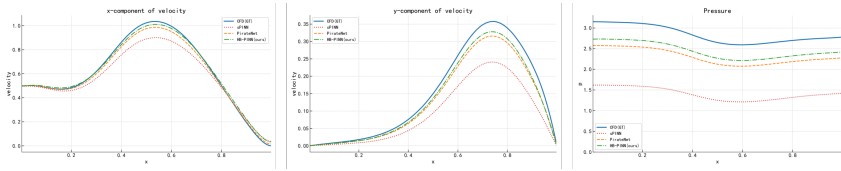

Figure 28: Comparison of velocity and pressure profiles along the $y = 0.5$ cross-section in Case 2.

Table 4: Comparative analysis of error metrics across different methods at multiple time steps

| Time | Method | Error Metrics | | |
|---|---|---|---|---|
| | | MSE($\downarrow$) | MAE($\downarrow$) | Relative L2($\downarrow$) |
| $t = 0.1$ | sPINN | 0.111 93 | 0.211 71 | 0.782 71 |
| | hPINN | 0.160 37 | 0.258 55 | 0.936 88 |
| | MFN-PINN | 0.105 44 | 0.206 14 | 0.759 67 |
| | XPINN | 0.054 66 | 0.151 06 | 0.546 99 |
| | SA-PINN | 0.076 38 | 0.172 18 | 0.646 59 |
| | PirateNet | 0.05228 | 0.14450 | 0.53493 |
| | HB-PINN | **0.00873** | **0.05070** | **0.21864** |
| $t = 0.3$ | sPINN | 0.119 06 | 0.216 25 | 0.788 15 |
| | hPINN | 0.154 41 | 0.241 52 | 0.897 54 |
| | MFN-PINN | 0.109 33 | 0.208 13 | 0.755 23 |
| | XPINN | 0.041 74 | 0.122 88 | 0.466 66 |
| | SA-PINN | 0.067 36 | 0.152 83 | 0.592 80 |
| | PirateNet | 0.01989 | 0.09057 | 0.32215 |
| | HB-PINN | **0.00580** | **0.04194** | **0.17405** |
| $t = 0.5$ | sPINN | 0.122 18 | 0.218 31 | 0.789 21 |
| | hPINN | 0.155 47 | 0.240 54 | 0.890 26 |
| | MFN-PINN | 0.111 64 | 0.209 94 | 0.754 41 |
| | XPINN | 0.040 46 | 0.119 80 | 0.454 16 |
| | SA-PINN | 0.077 00 | 0.164 65 | 0.626 53 |
| | PirateNet | 0.02027 | 0.08035 | 0.32152 |
| | HB-PINN | **0.00700** | **0.04895** | **0.18901** |
| $t = 0.7$ | sPINN | 0.123 68 | 0.219 27 | 0.789 48 |
| | hPINN | 0.156 07 | 0.240 33 | 0.886 85 |
| | MFN-PINN | 0.112 83 | 0.210 61 | 0.754 05 |
| | XPINN | 0.040 87 | 0.119 90 | 0.453 83 |
| | SA-PINN | 0.089 56 | 0.179 47 | 0.671 81 |
| | PirateNet | 0.02522 | 0.09204 | 0.35652 |
| | HB-PINN | **0.00805** | **0.05228** | **0.20079** |
| $t = 1.0$ | sPINN | 0.124 00 | 0.218 76 | 0.788 19 |
| | hPINN | 0.155 59 | 0.239 51 | 0.882 88 |
| | MFN-PINN | 0.105 43 | 0.201 50 | 0.726 78 |
| | XPINN | 0.041 49 | 0.120 28 | 0.455 92 |
| | SA-PINN | 0.106 12 | 0.196 42 | 0.729 15 |
| | PirateNet | 0.03202 | 0.10682 | 0.40054 |
| | HB-PINN | **0.00825** | **0.04870** | **0.20332** |

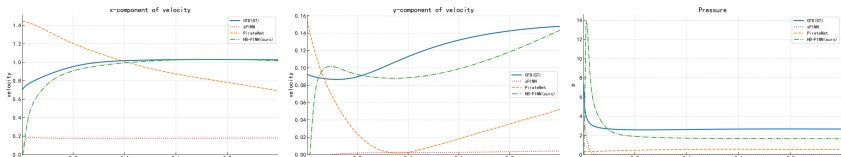

Figure 29: Comparison of Temporal Distributions of Velocity and Pressure at (0.5, 0.5) in Case 3.

# G   Steady-state flow in a square cavity with obstructed segmented inlet at different Reynolds numbers

As an extension of the comparative study on three flow models under the Reynolds number $Re = 100$ scenario in the main text, this supplementary investigation systematically evaluates model

performance at elevated Reynolds numbers ($Re = 500, 1000, 2000$) using the Case 2 configuration, while strictly maintaining the original geometric structure and boundary conditions.

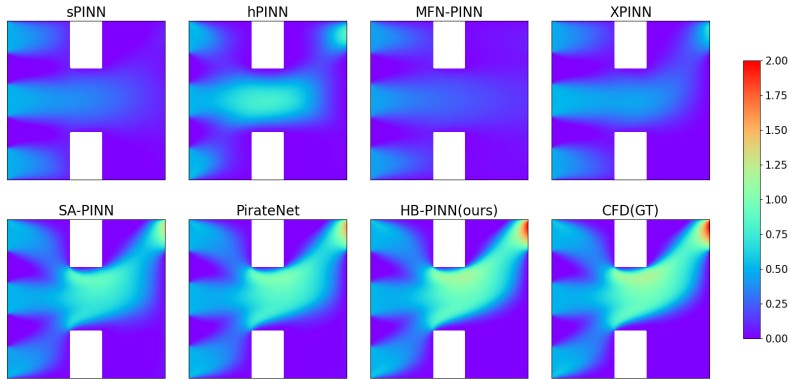

Figure 30: Comparison of velocity results under different methods for steady-state flow in a square cavity with obstructed segmented inlet at Reynolds number $Re = 500$.

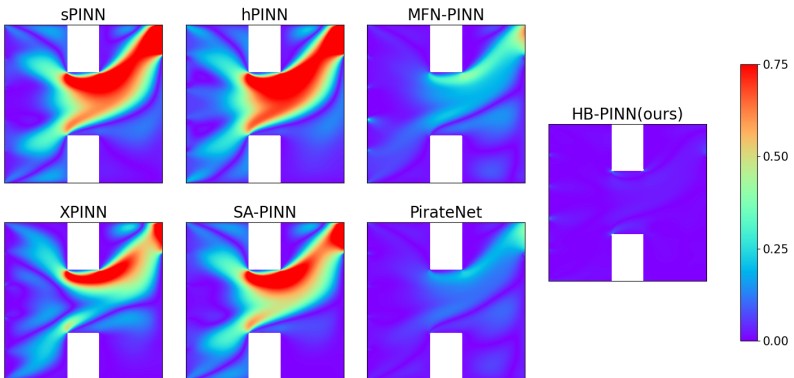

Figure 31: Comparison of velocity residuals relative to ground truth (GT) for steady-state flow in a square cavity with obstructed segmented inlet at $Re = 500$, evaluated under different numerical methods.

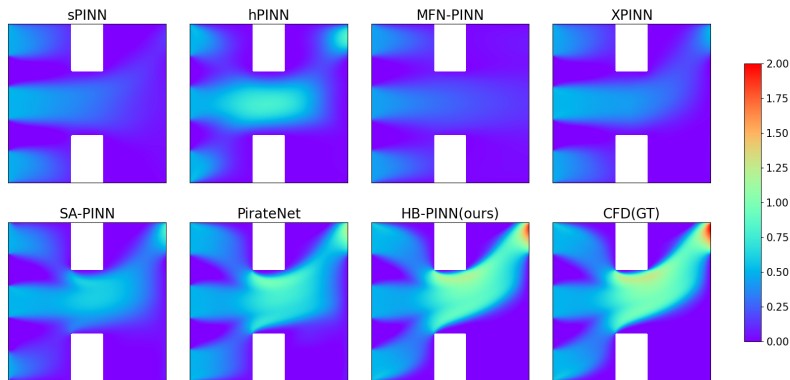

Figure 32: Comparison of velocity results under different methods for steady-state flow in a square cavity with obstructed segmented inlet at Reynolds number $Re = 1000$.

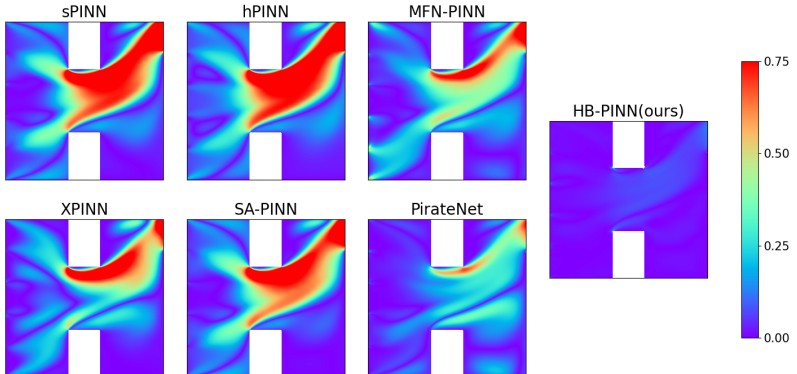

Figure 33: Comparison of velocity residuals relative to ground truth (GT) for steady-state flow in a square cavity with obstructed segmented inlet at $Re = 1000$, evaluated under different numerical methods.

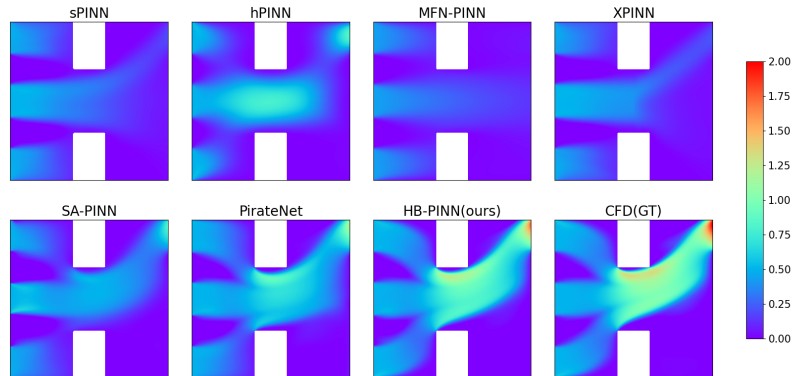

Figure 34: Comparison of velocity results under different methods for steady-state flow in a square cavity with obstructed segmented inlet at Reynolds number $Re = 2000$.

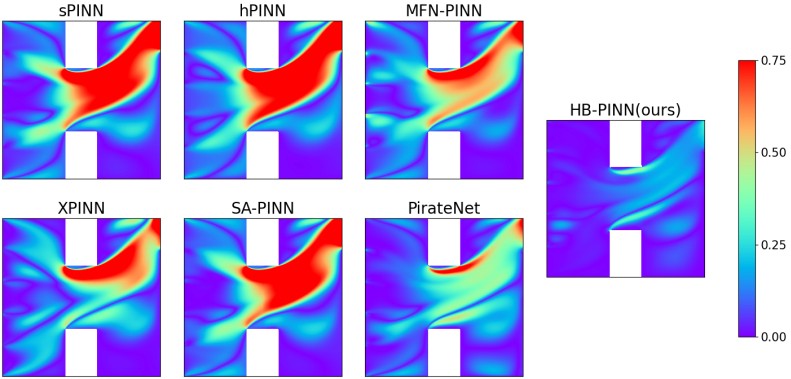

Figure 35: Comparison of velocity residuals relative to ground truth (GT) for steady-state flow in a square cavity with obstructed segmented inlet at $Re = 2000$, evaluated under different numerical methods.

Fig.30,32,34 present the velocity predictions of different methods at $Re = 500$, 1000, and 2000, respectively, while their residuals relative to high-fidelity CFD results are visualized in Fig.31,33,35. As shown in the MSE variation trends across Reynolds numbers (Fig.36), all methods exhibit rising

prediction errors with increasing $Re$. However, the proposed HB-PINN consistently achieves the lowest MSE values at every Reynolds number.

Quantitative error metrics under multiple evaluation criteria are tabulated in Table.5. Notably, HB-PINN achieves an order-of-magnitude reduction in MSE compared to conventional methods at $Re = 500$ and $1000$. Even under the most challenging $Re = 2000$ condition, HB-PINN demonstrates a notable accuracy improvement.

Table 5: Comparative analysis of error metrics across different methods under varying Reynolds numbers (Re).

| Re | Method | Error Metrics | | |
|---|---|---|---|---|
| | | MSE($\downarrow$) | MAE($\downarrow$) | Relative L2($\downarrow$) |
| 500 | sPINN | 0.114 63 | 0.205 69 | 0.740 24 |
| | hPINN | 0.056 82 | 0.139 10 | 0.521 16 |
| | MFN-PINN | 0.126 93 | 0.221 63 | 0.778 92 |
| | XPINN | 0.084 25 | 0.179 04 | 0.634 62 |
| | SA-PINN | 0.013 79 | 0.078 27 | 0.256 81 |
| | PirateNet | 0.00591 | 0.05145 | 0.16819 |
| | HB-PINN | **0.00071** | **0.01122** | **0.05865** |
| 1000 | sPINN | 0.125 41 | 0.215 97 | 0.753 06 |
| | hPINN | 0.068 73 | 0.156 08 | 0.557 50 |
| | MFN-PINN | 0.139 45 | 0.232 43 | 0.794 08 |
| | XPINN | 0.102 50 | 0.196 43 | 0.680 79 |
| | SA-PINN | 0.058 81 | 0.158 42 | 0.515 69 |
| | PirateNet | 0.02666 | 0.11171 | 0.34722 |
| | HB-PINN | **0.00364** | **0.02857** | **0.12837** |
| 2000 | sPINN | 0.133 26 | 0.226 16 | 0.755 87 |
| | hPINN | 0.080 26 | 0.170 93 | 0.586 63 |
| | MFN-PINN | 0.152 02 | 0.242 13 | 0.807 34 |
| | XPINN | 0.131 56 | 0.223 41 | 0.751 03 |
| | SA-PINN | 0.075 84 | 0.176 81 | 0.570 22 |
| | PirateNet | 0.04191 | 0.14184 | 0.42391 |
| | HB-PINN | **0.02075** | **0.06640** | **0.29832** |

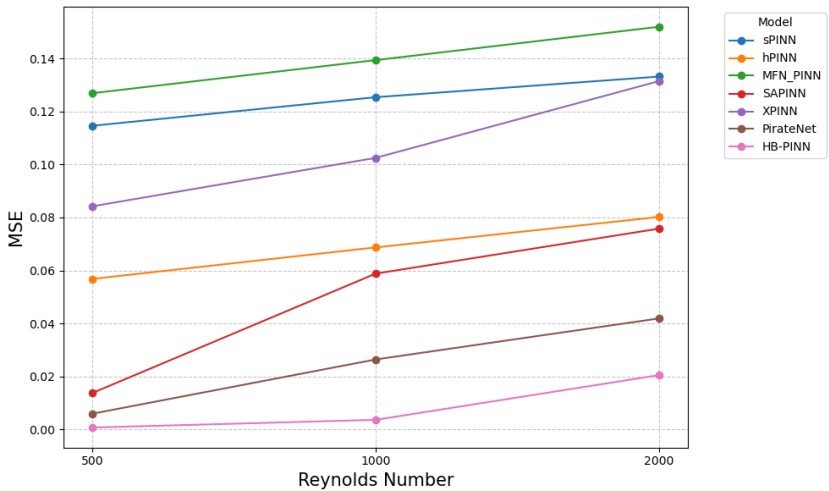

Figure 36: MSE Comparison with Reynolds Numbers

## H    The more complex obstructed cavity flow model

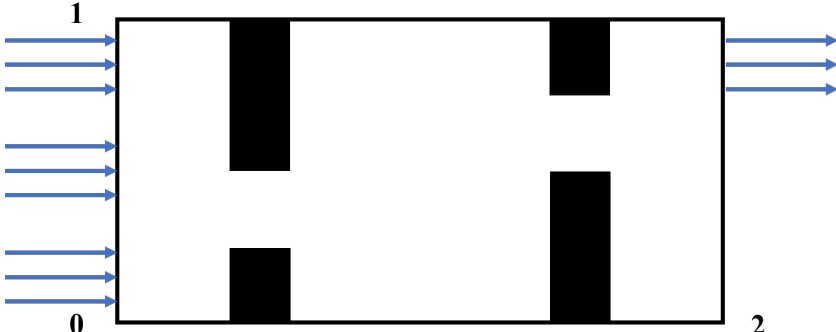

Figure 37: Extended schematic of the segmented inlet configuration in an obstructed rectangular cavity flow.

To validate the capability of our Hybrid Boundary Physics-Informed Neural Network (HB-PINN) in handling geometrically complex scenarios, we extend Case 2 from the main text to a more intricate configuration. As illustrated in Fig.37, the computational domain is expanded to a rectangular cavity of width 2 and height 1. Two types of staggered obstructions are embedded within the cavity:

- **Type A Obstructions:** Width $= 0.2$, Height $= 0.5$
- **Type B Obstructions:** Width $= 0.2$, Height $= 0.25$

The segmented inlet configuration (see Appendix E.1 for geometric specifics) imposes a normal inflow velocity $u = 0.5$, while the outlet, positioned at the upper-right corner of the cavity, enforces a static pressure $p = 0$. All other boundaries adhere to no-slip conditions ($u = v = 0$).

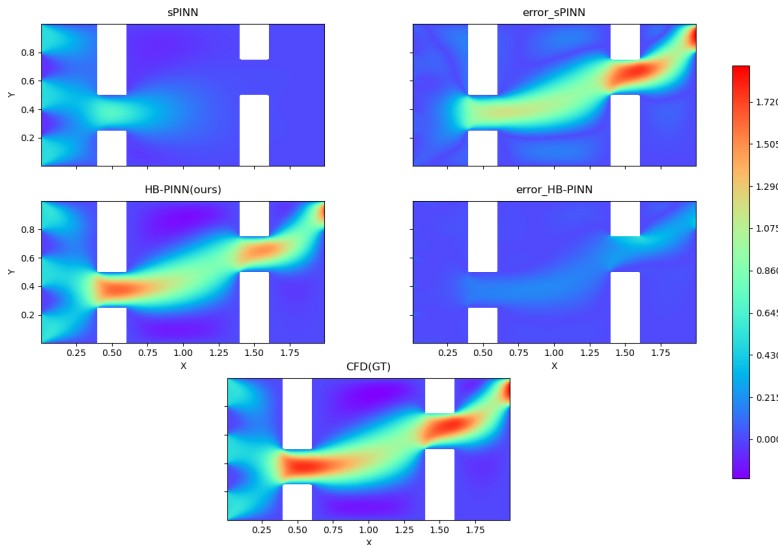

Figure 38: Velocity predictions and residuals of sPINN versus HB-PINN in the extended obstructed cavity flow model. Left: Velocity predictions; Right: Absolute errors relative to CFD; Bottom: CFD-computed results.

Fig.38 compares the velocity prediction results of conventional soft-constrained PINN (sPINN) and the proposed HB-PINN in the extended obstructed cavity flow model. As the geometric and physical

complexity of the model increases, traditional PINN methods struggle to resolve conflicts between competing loss terms (e.g., boundary condition residuals vs. governing equation residuals), leading to significant performance degradation in velocity field predictions. In contrast, HB-PINN maintains robust training convergence and achieves consistently accurate results. More detailed error metrics are summarized in the "Complex Cavity Flow" section of Table 6.

## I  Heat equation

To demonstrate the applicability of the HB-PINN method to other PDE systems, we evaluate its performance on a two-dimensional transient heat conduction problem, governed by the following equation:

$$\frac{\partial T}{\partial t} = \alpha \left( \frac{\partial^2 T}{\partial x^2} + \frac{\partial^2 T}{\partial y^2} \right), \tag{12}$$

$T(x, y, t)$ represents the temperature field, where the thermal diffusivity $\alpha = \frac{k}{\rho c}$ is defined by the thermal conductivity $k$, density $\rho$, and specific heat capacity $c$. The geometric configuration, shown in Fig.39, features a square domain with edge length $1$, containing four square heat sources (edge length $0.1$) centered at coordinates $(0.3, 0.3)$, $(0.3, 0.7)$, $(0.7, 0.3)$, and $(0.7, 0.7)$. These heat sources are maintained at a dimensionless temperature of $1$, while all external boundaries are fixed at $0$.

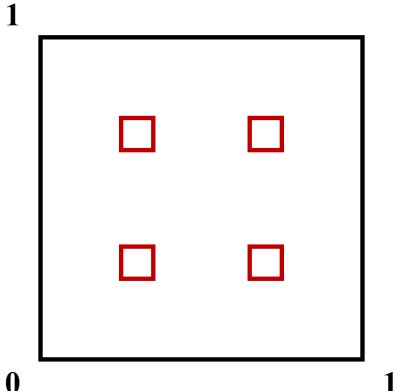

Figure 39: Geometric configurations employed in heat conduction problems

## J  Lid-driven Cavity flow

In this section, we evaluate the performance of HB-PINN in lid-driven cavity (LDC) flow, a classical benchmark problem in computational fluid dynamics. The LDC flow involves simulating the motion of an incompressible fluid within a two-dimensional square cavity, where the top lid is assigned a horizontal velocity of $u = 1$, while the other walls maintain a no-slip condition. The boundary conditions are mathematically formulated as follows:

$$u = 1, \quad 0 \leq x \leq 1, \ y = 1; \qquad u = 0, \quad \text{others};$$
$$v = 0, \quad \text{on } \partial\Omega; \qquad\qquad p = 0, \quad (0, 0).$$

The velocity predictions of sPINN and HB-PINN are shown in Fig.42. Due to the simpler geometric configuration of the model compared to the cases discussed in the main text, the improvement in accuracy is less pronounced than that observed in complex-boundary models. However, as evidenced by the residual results in Fig.43, HB-PINN exhibits significantly lower errors near boundaries. This finding aligns with the conclusions drawn in Appendix G, demonstrating that HB-PINN consistently achieves higher precision in boundary regions across diverse models. More detailed error metrics are summarized in the "LDC" section of Table 6.

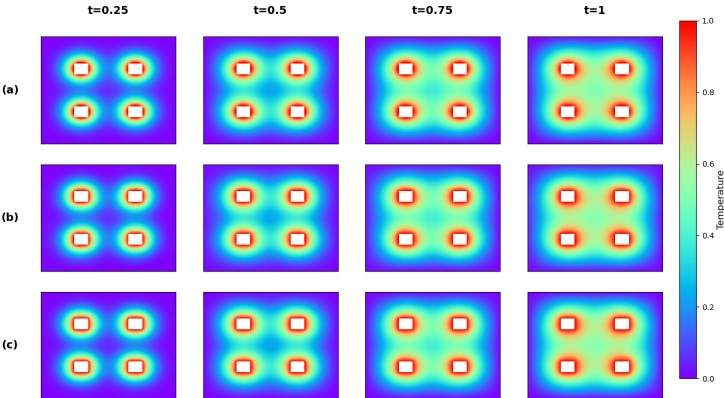

Figure 40: Comparison of temperature results at $t = 0.25, 0.5, 0.75$, and $1.0$ for the heat conduction problem: (a) sPINN predictions; (b) HB-PINN predictions; (c) ground truth (GT).

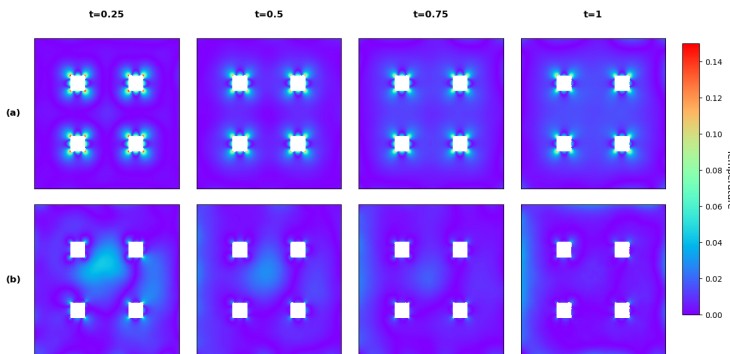

Figure 41: Residuals between predicted temperature and ground truth (GT) at $t = 0.25, 0.5, 0.75$, and $1.0$ for the heat conduction problem: (a) Residuals of sPINN predictions;(b) Residuals of HB-PINN predictions.

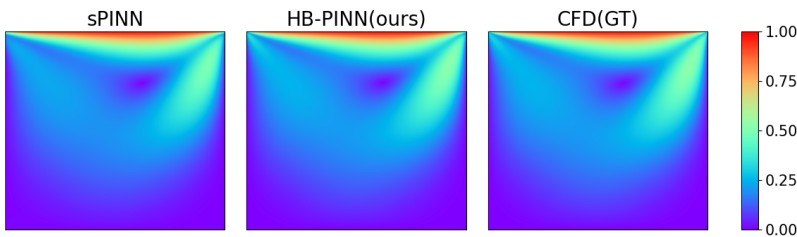

Figure 42: Comparison of Velocity Predictions Between sPINN, HB-PINN, and CFD in Lid-Driven Cavity Flow.

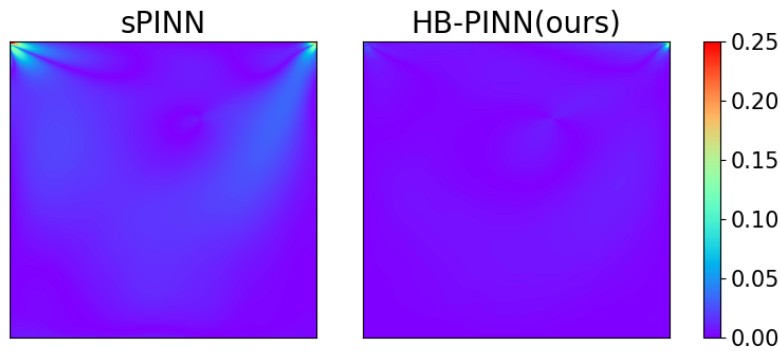

Figure 43: Residuals of Velocity Predictions Between sPINN, HB-PINN, and CFD in Lid-Driven Cavity Flow

Table 6: Comparative analysis of error metrics across the more complex obstructed cavity flow, heat equation, and Lid-Driven Cavity (LDC) flow

| Problem | Method | Error Metrics | | |
|---|---|---|---|---|
| | | MSE($\downarrow$) | MAE($\downarrow$) | Relative L2($\downarrow$) |
| Complex Cavity Flow | sPINN | 0.215 89 | 0.257 29 | 0.838 66 |
| | HB-PINN | **0.01331** | **0.05637** | **0.20829** |
| Heat equation ($t = 1$) | sPINN | 0.002 76 | 0.017 80 | 0.107 73 |
| | HB-PINN | **0.00090** | **0.01071** | **0.06152** |
| LDC | sPINN | 0.000 23 | 0.011 52 | 0.058 73 |
| | HB-PINN | **0.00002** | **0.00377** | **0.02001** |

## K   Training and Inference Times

Table 7: Comparison of Training and Inference Times

| Method | Training Time (1000 epochs) | Inference Time |
|---|---|---|
| SPINN | 31.1 s | – |
| hPINN | 48.3 s | – |
| MFN-PINN | 92.5 s | – |
| XPINN | 101.5 s | – |
| SA-PINN | 41.4 s | – |
| PirateNet | 106.9 s | – |
| HB-PINN | **47.8 s** | **0.29 s** |
| CFD | – | 17 s |

