# OpenReview forum: "Hybrid Boundary Physics-Informed Neural Networks for Solving Navier-Stokes Equations with Complex Boundary"
_NeurIPS.cc/2025/Conference — NeurIPS 2025 poster_

### Official Review · Reviewer_qJeV · 2025-06-28

**Clarity:** 3
**Significance:** 3
**Originality:** 2
**Rating:** 5
**Confidence:** 3

**Summary:**

The paper introduces Hybrid Boundary Physics-Informed Neural Networks (HB-PINN), a novel approach to solving Navier-Stokes equations (NSEs) with complex boundary conditions. The method addresses the limitations of traditional PINNs, which struggle to balance boundary condition losses and PDE residual losses, especially in geometrically complex domains. HB-PINN achieves lower errors compared to baseline PINN approaches (e.g., sPINN, hPINN, MFN-PINN, XPINN, SA-PINN, PirateNet). Extensive experiments and ablation studies demonstrate its robustness and accuracy in handling complex boundary conditions.

**Questions:**

1. How sensitive is the performance of HB-PINN to the choice of α in the distance metric network?
2. How does the training time and resource consumption of HB-PINN compare to baseline methods?
3. Can the HB-PINN framework be extended to 3D Navier-Stokes equations?
4. How does HB-PINN perform for geometries with moving boundaries or time-dependent boundary conditions?
5. Have you explored the application of HB-PINN to industrial-scale problems (e.g., aerodynamics, weather modeling)?
6. How does HB-PINN compare to traditional CFD solvers in terms of accuracy and efficiency for such problems?

**Ethical Concerns:**

["NO or VERY MINOR ethics concerns only"]

**Final Justification:**

The response mostly addressed my concerns, and I raised the score. Some issues remain to be resolved in future work, and I understand that these are common challenges faced by PINN-type methods, which may be difficult to overcome in the short term. At the very least, this paper has made effective attempts and progress in this field.

**Limitations:**

Yes

**Paper Formatting Concerns:**

No Formatting Concerns

**Quality:**

3

**Strengths And Weaknesses:**

Strengths
1. Decoupling subnetworks (NP, ND, NH) effectively handles complex boundaries. The distance metric network (ND) is a useful solution for handling complex geometries, ensuring smooth transitions near boundaries.
2. Error reductions compared to baseline methods. Detailed residual analyses and comparison plots provide clear evidence.
3. The methodology is extended to transient flows, high Reynolds number scenarios, and other PDEs (e.g., heat conduction). The results indicate potential applicability to a wide range of fluid dynamics problems and other scientific domains.

Weaknesses
1. Empirical Hyperparameter Selection:
The critical parameter α in the distance metric network and the number of pre-training epochs for NP are tuned empirically. This lack of systematic optimization may limit reproducibility and scalability to other problems. The paper does not provide guidance on how to set these parameters for new applications.
2. Computational Cost:
While the paper demonstrates improved accuracy, it does not discuss the computational overhead of training three separate subnetworks.
3. The paper primarily focuses on toy problems but does not discuss the practical implications and limitations of HB-PINN for real-world engineering problems (e.g., industrial CFD applications).

---

> ### Author Rebuttal · Authors · 2025-07-31
>
> **For Q1**
>
> **A1**: The performance of HB-PINN is indeed dependent on $\\alpha$, and either excessively large or small values of $\\alpha$ adversely affect training.
>
> If $\\alpha$ is too small (e.g., $\\alpha=1$), the weight of the PDE loss in the interior domain becomes insufficient. Conversely, if $\\alpha$ is too large (e.g., $\\alpha=20$), the boundary layer becomes excessively thin. Both scenarios impede effective training of the main network. We conducted relevant ablation studies in Appendix C.3. Generally, when $\\alpha$ falls within the range of 5–10, the final training results of the main network vary minimally. Therefore, selecting $\\alpha$ within the 5–10 range is optimal.
>
> Based on observations and repeated experiments across multiple different models, we can provide the following recommendations for the selection ranges of the key parameters $\\alpha$ and the number of pre-training iterations for the $\\mathcal{N}\_{\\mathcal{P}}$ network:
>
> 1.The distance metric network parameter $\\alpha$ can be chosen between 5 and 10, adjusted according to the density of obstacles (i.e., the distance between different boundaries) within the domain. Sparser obstacle distributions correspond to smaller $\\alpha$  values.
>
> 2.The number of pre-training iterations for the $\\mathcal{N}\_{\\mathcal{P}}$ network can be chosen between 10,000 and 50,000, adjusted based on the complexity of the boundary conditions. Higher complexity warrants selecting a higher number of iterations to ensure strict satisfaction of the boundary conditions.
>
> However, we have not yet been able to summarize an adaptive adjustment strategy for $\\alpha$ and the $\\mathcal{N}\_{\\mathcal{P}}$ pre-training iteration count that suits all scenarios. In future work, we will continue research on hyperparameter tuning to develop adaptive parameter selection schemes applicable to different situations.
>
>
> **For Q2**
>
> **A2**: We thank the reviewer for their interest in the computational efficiency characteristics of HB-PINN. Table 1 details the comparative training and inference time costs for different PINN methods on Case 2 under unified experimental settings (training on an NVIDIA RTX 4090 GPU, CFD run on an Intel Xeon E5-2640 v4 CPU). All models uniformly accept inputs with a shape of $[20000, 2]$.
>
> **Training Efficiency:**
>
> The training time for HB-PINN (47.8 s) is significantly better than most advanced PINN variants (MFN-PINN: 92.5 s, XPINN: 101.5 s, PirateNet: 106.9 s). It is comparable to hPINN (48.3 s) and SA-PINN (41.4 s), and only slightly longer than the most basic sPINN (31.1 s). The training speed of certain methods (such as MFN and PirateNet) is reduced due to complex mathematical operations beyond the model architecture itself. Critically, while maintaining training costs on par with advanced methods, HB-PINN achieved significantly lower prediction errors, achieving an optimized balance between accuracy and efficiency.
>
> **Table 1:** Performance comparison of different PINN methods
>
> $$
> \\begin{array}{l c c c c c c c}
> \\hline
>  & \\text{SPINN} & \\text{hPINN} & \\text{MFN-PINN} & \\text{XPINN} & \\text{SA-PINN} & \\text{Pirate Net} & \\text{HB-PINN} \\\\
> \\hline
> \\text{Training time (s)} & 31.1 & 48.3 & 92.5 & 101.5 & 41.4 & 106.9 & 47.8 \\\\
> \\text{Params (M)} & 0.30 & 0.34 & 0.36 & 0.16 & 0.30 & 0.06 & 0.36 \\\\
> \\text{FLOPs (G)} & 5.92 & 6.65 & 7.03 & 1.49 & 5.92 & 1.73 & 7.11 \\\\
> \\hline
> \\end{array}
> $$
>
> **For Q3**
>
> **A3**: We thank the reviewer for raising this crucial question concerning the universality of the method. Solving three-dimensional (3D) transient flow models under more realistic conditions is both a necessary direction for PINN development and a formidable challenge.
>
> The HB-PINN framework is inherently capable of extension to 3D. We have currently tested it on a relatively simple 3D model (3D steady-state cavity flow), achieving an error range of $10^{-3}$ to $10^{-4}$ across multiple calculated 2D cross-sections. However, due to time constraints, more detailed comparisons and testing on complex 3D transient cases have not yet been conducted. Solving more complex and realistic problems remains a focus for future research.
>
> To support this goal, we have established a laboratory setup for generating flow fields to conduct complementary physics-based experiments. In future work, we plan to cross-validate our computational results with real experimental data to better address the challenges associated with three-dimensional flow problems.
>
>
>
> **For Q4**
>
> **A4**: We sincerely thank the reviewer for highlighting the importance of addressing moving boundaries and time-varying boundary conditions, which are indeed representative of many real-world scenarios and hold significant research value.
>
> Although our current work does not directly address these challenges, we believe that HB-PINN has inherent potential to handle moving or time-varying boundary conditions for the following reasons:
>
> 1.The $\\mathcal{N}\_{\\mathcal{P}}$ network incorporates weak constraints derived from the PDE. This enables it to provide a physically reasonable initialization for the main network’s training across diverse boundary conditions, reducing sensitivity to temporal variations.
>
> 2.The main network ($\\mathcal{N}\_{\\mathcal{H}}$) focuses solely on satisfying the governing equations. This design minimizes interference from loss terms related to time-varying boundaries.
>
> Therefore, we believe HB-PINN possesses inherent potential to handle moving or time-varying boundary conditions. Future work will explicitly investigate these challenging scenarios.
>
>
>
>
> **For Q5**
>
> **A5**: We appreciate the reviewer’s interest in the broader applicability of HB-PINN. As mentioned in Q3\A3, we have established a dedicated laboratory setup capable of generating controlled flow fields at a $1\\,\\text{m} \\times 1\\,\\text{m} \\times 1\\,\\text{m}$ scale to simulate atmospheric phenomena. However, there are still technical challenges in accurately capturing flow velocity. We are actively working to address these issues and plan to evaluate our method under real-world experimental conditions in the near future. This will serve as an important first step toward extending HB-PINN to industrial-scale applications.
>
>
>
>
>
> **For Q6**
>
> **A6**: At the current stage of our research, we primarily use CFD simulation results as the benchmark (Ground Truth) to evaluate the accuracy of HB-PINN compared to other PINN methods. This approach is widely adopted in PINN research due to the technical challenges associated with directly validating physics-informed neural networks against real experimental data.
>
> Therefore, we are currently only able to compare efficiency against CFD solvers. When predicting results on a $200*200$ grid, the trained PINN method achieves inference times on the millisecond scale, whereas the CFD method requires 17 seconds(as Table2). This orders-of-magnitude advantage endows HB-PINN with greater potential for real-time simulation, parameter scanning, and optimal control – scenarios demanding high-frequency access to flow field data.
>
> **Table 2:** Inference time comparison of different methods (in seconds)
>
> $$
> \\begin{array}{l c c c c c c c c}
> \\hline
>  & \\text{sPINN} & \\text{hPINN} & \\text{MFN-PINN} & \\text{XPINN} & \\text{SA-PINN} & \\text{Pirate Net} & \\text{HB-PINN} & \\text{CFD} \\\\
> \\hline
> \\text{Inference time} & 0.075 & 0.052 & 0.051 & 0.029 & 0.034 & 0.027 & 0.042 & 17.000 \\\\
> \\hline
> \\end{array}
> $$
>
>
> Regarding the accuracy comparison with CFD solvers, we are eager to conduct such tests once our laboratory setup is fully operational.

---

> > ### Comment · Reviewer_qJeV · 2025-08-07
> >
> > Thank you for the response. It mostly addressed my concerns, and I raised the score. Some issues remain to be resolved in future work, and I understand that these are common challenges faced by PINN-type methods, which may be difficult to overcome in the short term. At the very least, this paper has made effective attempts and progress in this field.

---

> > > ### Author Response · Authors · 2025-08-07
> > >
> > > We thank you for reviewing our response and for raising your rating. We deeply appreciate your feedback and are pleased that our response has addressed most of your concerns. We will continue researching to tackle more complex and realistic problems in the future. Thank you again for your constructive comments and your recognition of the manuscript's progress.

---

### Official Review · Reviewer_Zovt · 2025-07-01

**Clarity:** 2
**Significance:** 2
**Originality:** 2
**Rating:** 4
**Confidence:** 3

**Summary:**

A kind of mixture model is proposed for dealing with complex boundary conditions with PINNs. The idea is to (softly) switch between two networks: one predominantly for achieving boundary conditions and another for the inner domain solution. The method is examined with multiple different flows.

**Questions:**

Points **(2)**, **(3)**, and **(5)** in the "Strengths And Weaknesses" section  are my questions particularly affecting the evaluation.

**Ethical Concerns:**

["NO or VERY MINOR ethics concerns only"]

**Final Justification:**

The response mostly addressed my concerns, and thus I raised the rating. With that being said I cannot assess how the paper can be improved to make the contribution clearer (as done in the response), not only to me but to the general audience, thus my rating remains around borderline.

**Limitations:**

Yes.

**Quality:**

3

**Strengths And Weaknesses:**

### Strengths

- The motivation to separately deal with the boundary and inner regions is clearly presented.
- The proposed method is compared with multiple baseline methods.

### Weaknesses

**(1)**
Despite the (ostensible) resemblance of the proposed method to hPINN by Lu et al. (2021) [21], I do not think it is discussed well.
The design choices different from hPINN should be elaborated on more, with the intentions behind them explained.
For example, why $\mathcal{N}\_\mathcal{P}$ is used instead of the original boundary condition function? Why $\mathcal{N}_\mathcal{D}$ is necessary? etc.

**(2)**
The motivation to use $\mathcal{N}\_\mathcal{D}$ is particularly unclear.
What is the point of learning $\mathcal{N}_\mathcal{D}$ when you can compute $\hat{\mathcal{D}}$?
Why not using $\hat{\mathcal{D}}$ in Eq. (3) directly?

**(3)**
In the loss function for the particular solution net $\mathcal{N}\_\mathcal{P}$, the motivation to have non-zero $\lambda_1$ is unclear. Setting much smaller $\lambda_1$ than $\lambda_2$ and $\lambda_3$ is proposed, which itself makes sense, but why $\lambda_1>0$ is preferable? Is there any ablation study of setting $\lambda_1=0$?

**(4)**
Line 172:

> hPINN method in this study selectively relaxes the enforcement of boundary conditions

The exact setup of the hPINN here is unclear. Please elaborate on the way the BC was relaxed.

**(5)**
Perhaps related to (4) above, I am not sure if an important baseline (or ablated) method is examined. The proposed method should be compared with a variant of hPINN, where the $\ell$ function of hPINN is replaced with the proposed $\hat{\mathcal{D}}$ function (with the other components being the same with the original hPINN; i.e., no $\mathcal{N}_\mathcal{P}$, and no $\mathcal{N}\_\mathcal{D}$). Is there any such baseline or ablated method?

**(6)**
The details of the other baseline methods are missing as well. Please clarify the configurations of all the baseline methods.

---

> ### Author Rebuttal · Authors · 2025-07-31
>
> **For Q1&2(W1&2)**
>
> **A1&2**: We thank the reviewer for their valuable comments. These questions help us clarify the motivation behind our method's design and its differences from hPINN more clearly. We fully agree with the reviewer's point regarding the importance of detailing our design choices and their underlying rationale.
>
> **1. Why use $\\mathcal{N}\_{\\mathcal{P}}$ instead of the original boundary condition function?**
>
> Constraining the solution network $\\mathcal{N}\_{\\mathcal{P}}$ solely with boundary conditions works when the boundary structure is simple. However, for complex boundaries, this approach leads to distorted or discontinuous outputs within the solution domain, especially near junctions between different boundary types. This significantly increases the training difficulty of the main network, even hindering convergence – as we observed in our hPINN experiments: strict constraints on certain boundary regions (e.g., wall regions near the inlet, more details described in Q4) had to be relaxed for the main network loss to decrease effectively. Otherwise, the loss became extremely large and failed to converge, making it impossible to obtain valid comparison results (as shown by the initial loss descent trend in Table 1).
>
> However, this compromise violates the requirement for full boundary condition satisfaction, which is unacceptable for solving physical models. This precisely highlights the dilemma of traditional hard-constraint methods when handling complex boundary problems. Therefore, we introduced $\\mathcal{N}\_{\\mathcal{P}}$ as an alternative solution aimed at more robustly handling complex boundary constraints.
>
> **2. What is the rationale for the necessity of $\\mathcal{N}\_{\\mathcal{D}}$? What is the significance of training the $\\mathcal{N}\_{\\mathcal{D}}$ network when an analytical distance $\\hat{\\mathcal{D}}\_{q}$ can be directly computed? Why not use the analytical $\\hat{\\mathcal{D}}\_{q}$ directly in Equation (3)?**
>
> For geometrically simple models (like the vascular model mentioned in Sun et al. (2020) [11]), analytical expressions for the distance can indeed be easily obtained. However, for most complex geometries, computing the distance typically requires calculating distances to individual boundary segments and then taking the minimum. Crucially, this minimum operation is non-differentiable, preventing its integration with the main network for joint backpropagation. Even if approximate expressions like the R-function can be used, constructing the corresponding approximate function representation for complex geometries remains highly complex (we believe its construction difficulty is even greater than training a distance function using the $\\mathcal{N}\_{\\mathcal{D}}$ network). More importantly, compared to using the distance network, analytical/approximate methods significantly increase the time cost of training the main network (approximately 2:1). This is because the neural network's inherent auto-differentiation capability enables seamless integration with the main PINN network, ensuring efficiency during joint training and gradient backpropagation. Based on these considerations, we chose the more convenient and efficient $\\mathcal{N}\_{\\mathcal{D}}$ network to construct the distance function. This is a design that balances universality and efficiency for handling complex geometries.
>
> **Table 1:** Loss comparison across different methods at various epochs
>
> $$
> \\begin{array}{c c c c}
> \\hline
> \\text{Epoch} & \\text{Original hPINN} & \\text{Relaxed hPINN} & \\text{HB-PINN} \\\\
> \\hline
> 0     & 2335810 & 47.44 & 7.129 \\\\
> 5000  & 3318232 & 26.81 & 1.244 \\\\
> 10000  & 5633070 & 25.59 & 1.070 \\\\
> 15000  & 1076081 & 11.20 & 0.810 \\\\
> \\hline
> \\end{array}
> $$
>
> **For Q3(W3)**
>
> **A3: Regarding the Motivation for $\\lambda\_1 > 0$:**
>
> The core motivation for setting $\\lambda\_1$ to a non-zero value (albeit much smaller than $\\lambda\_2$ and $\\lambda\_3$) in the loss function of the particular solution network $\\mathcal{N}\_{\\mathcal{P}}$ is to introduce weak constraints derived from the PDE. This design aims to:
>
> **1.Mitigate Internal Irregularity:** Constraining $\\mathcal{N}\_{\\mathcal{P}}$ solely with boundary conditions (i.e., $\\lambda\_1=0$) often leads to uncontrolled (irregular or distorted) solutions within the solution domain interior, particularly away from boundaries.
>
> **2.Reduce Main Network Training Difficulty:** Imposing preliminary weak PDE constraints during training significantly increases the controllability of the solution produced by the $\\mathcal{N}\_{\\mathcal{P}}$ network. This results in an initial solution from $\\mathcal{N}\_{\\mathcal{P}}$ that is closer to the true physical field, providing a better starting point for the subsequent training of the main network. This effectively reduces the initial training difficulty for the main network (this is closely related to the dilemma of handling complex boundaries with boundary constraints alone, as revealed in our responses to A1&2). The beneficial effect of $\\lambda\_1 > 0$ on initial loss is reflected in the HB-PINN loss components documented in Table 1.
>
> **Regarding the Ablation Study:**
>
> We fully agree with the reviewer on the importance of ablation studies for validation and have presented detailed results in Appendix C.1. Specifically:
>
> 1.The mD configuration in the experiment corresponds to the case where $\\lambda\_1 = 0$.
>
> 2.The error results in Appendix Table 2 clearly show that setting $\\lambda\_1 > 0$ reduces the solution error by approximately an order of magnitude compared to $\\lambda\_1 = 0$ (mD). It is crucial to emphasize that this significant advantage was achieved under conditions where we appropriately relaxed constraints on certain boundary segments (relevant details will be elaborated in Q4).
>
> 3.Furthermore, the loss records from the experiments provide additional evidence: $\\lambda\_1 = 0$ results in a poor-quality solution from the trained $\\mathcal{N}\_{\\mathcal{P}}$ network, which is severely detrimental to the subsequent training of the main network (specific data can be referenced in the original hPINN loss records in Table1).
>
> **For Q4(W4)**
>
> **A4**: We thank the reviewer for their attention to the rigor of experimental details and for highlighting the importance of clearly explaining the boundary condition relaxation operation.
>
> In the hPINN comparative experiments in this paper (including the mD configuration in ablation study C.1), we selectively relaxed the strict constraints on the wall boundaries near the inlet **(Due to the uncontrolled solutions generated within the solution domain, as mentioned in A1,2, and A3)**.
>
>  To enable the hPINN method to produce a trainable solution, we proactively reduced the enforcement strength of the wall constraints in the conflict region near the inlet (specifically, $0 < x < 0.4, y=0$ \& $y=1$ in Case 2). This is evident in the error plot for the mD configuration in Fig. 13 of Appendix C.1, where significant errors near the upper and lower walls close to the inlet are observable, precisely resulting from the relaxation of the relevant wall constraints.
>
> It is crucial to reiterate: While this operation allows the main network loss to decrease and seemingly yields results closer to Ground Truth, it is fundamentally an expedient measure that violates physical reality.
>
> We will add more detailed figures in the appendix illustrating the reasons for relaxing these constraints.
>
> **For Q5(W5)**
>
> **A5**: We thank the reviewer for raising this important comparison suggestion, which helps more comprehensively evaluate the contribution of the method's components. The improvement strategy you inquired about—namely, separately removing the $\\mathcal{N}\_{\\mathcal{P}}$ and $\\mathcal{N}\_{\\mathcal{D}}$ modules—has been validated in the ablation study presented in Appendix C.1.
>
> In Appendix C.1, we designed two configurations, mD and mP, for comparison:
>
> **1.mD Configuration:** This configuration used only the distance function network $\\mathcal{N}\_{\\mathcal{D}}$ and did not use the particular solution network $\\mathcal{N}\_{\\mathcal{P}}$. For the same reason explained in the previous Q4 response (i.e., the intense physical conflict at the wall boundaries near the inlet), the mD configuration also required relaxing the boundary constraints in that region to ensure trainability.
>
> **2. mP Configuration:** This configuration used only the particular solution network $\\mathcal{N}\_{\\mathcal{P}}$ and did not use the distance function network $\\mathcal{N}\_{\\mathcal{D}}$. The experimental results show that the solution errors for both the mD (only $\\mathcal{N}\_{\\mathcal{D}}$) and mP (only $\\mathcal{N}\_{\\mathcal{P}}$) configurations were significantly lower than those of the original hPINN method. Furthermore, the error of the complete HB-PINN model, which integrates both $\\mathcal{N}\_{\\mathcal{P}}$ and $\\mathcal{N}\_{\\mathcal{D}}$, was further significantly reduced. This demonstrates that both our proposed $\\mathcal{N}\_{\\mathcal{P}}$ and $\\mathcal{N}\_{\\mathcal{D}}$ networks played a positive role in improving the model's accuracy.
>
>
>
> **For Q6(W6)**
>
> **A6**: We thank the reviewer for their attention to the reproducibility of the experiments. All baseline methods maintain consistency in the following core parameters:
>
> **Table 2:** Uniform configuration for all baseline methods
>
> $$
> \\begin{array}{lc}
> \\hline
> \\text{Parameter Category} & \\text{Unified Configuration} \\\\
> \\hline
> \\text{Network Architecture} & 6\\ \\text{hidden layers} \\times 128\\ \\text{neurons} \\\\
> \\text{Training Iterations} & 3\\times10^{5} \\\\
> \\text{Optimizer} & \\text{Adam} \\\\
> \\text{Initial Learning Rate} & 1\\times10^{-3} \\\\
> \\text{Points per Epoch} & 20,000\\ \\text{(random sampling from domain)} \\\\
> \\hline
> \\end{array}
> $$
>
> We will add more detailed configuration details for the other baseline methods in the appendix.

---

> > ### Comment · Reviewer_Zovt · 2025-08-02
> >
> > Thank you for the response. It mostly addressed my concerns, and thus I raised the rating. With that being said I cannot assess how the paper can be improved to make the contribution clearer (as done in the response), not only to me but to the general audience, thus my rating remains around borderline.
> >
> > On **A1**:
> > Thanks for the explanation, I think I understood the point. It should be much more emphasized in the paper. Some concrete examples of the "extremely large" loss values, as shown in your new Table 1, will be highly helpful too. Hopefully some more detailed analyses (not only the qualitative explanation in the response) of why such large loss values may happen.
> >
> > On **A2**:
> > As the output of $\mathcal{N}_D$ is just multiplied to the output of $\mathcal{N}_H$, the gradients of the parameters of the other two networks, $\mathcal{N}_P$ and $\mathcal{N}_H$, can be computed even if $\mathcal{N}_D$ is not differentiable. So the motivation based on the need for backpropagation does not make much sense.
> > Meanwhile, the motivation to make the computation of the distance faster, as a kind of interpolator, might be reasonable, though I don't have sufficient evidence to fully justify it.
> > Probably the best way to show the need for $\mathcal{N}_D$ is by presenting the empirical results, probably those deferred to in the appendix now, in a more effective way.
> >
> > On **A5**:
> > Thank you for pointing out the materials in the appendix. I would suggest moving some of them to the main text, especially to emphasize how the proposed method's particular structure was helpful compared to existing methods and partial modifications.

---

> ### Author Response · Authors · 2025-08-02
>
> Thank you for your thoughtful response and for raising the rating. We appreciate your recognition of the improvements made in addressing your concerns. Regarding your comment on making the contribution clearer to a broader audience, we fully acknowledge the importance of clearly communicating the core ideas of our work. We will carefully incorporate your feedback and make further efforts to improve the clarity and accessibility of the paper.
>
>
> **For “On A1”:**
>
> Thank you for your suggestion. To further clarify the "extremely large" loss values, which are critical for demonstrating the significance of HB-PINN, we will provide deeper insights in the paper, supported by visualization results of the solution network $\\mathcal{N}\_{\\mathcal{P}}$ constrained solely by boundary conditions (as noted in A1/A2, this can lead to uncontrolled behavior within the solution domain). Additionally, we will supplement the appendix with loss landscape visualizations that directly reflect the data presented in the new Table 1 to facilitate further discussion.
>
> **For “On A2”:**
>
> **1.Regarding computational efficiency:** Your insight is correct — the computational acceleration capability is indeed a key reason for adopting the $\\mathcal{N}\_{\\mathcal{D}}$ network. To support this, we provide a concise comparison of the time costs for computing the distance field in our main Case 2 scenario, using the following three methods:
>
> (1) Direct minimum distance calculation (non-differentiable)
>
> (2) Neural network approximation via $\\mathcal{N}\_{\\mathcal{D}}$
>
> (3) R-function–based approximate distance function[25]
>
> Note: All timing results exclude backpropagation
>
> The results of Table3 demonstrate the significant speed advantage of the $\\mathcal{N}\_{\\mathcal{D}}$ network.
>
>
> **Table 3:** Computation time comparison of different methods (for 1000 iterations)
>
> $$
> \\begin{array}{l c c c}
> \\hline
>  & \\text{Direct computation} & \\text{$\\mathcal{N}\_{\\mathcal{D}}$ network} & \\text{R-function} \\\\
>  & \\text{(non-differentiable)} & & \\\\
> \\hline
> \\text{Computation time} & 2.43\\,\\text{s} & 0.77\\,\\text{s} & 909.89\\,\\text{s} \\\\
> \\hline
> \\end{array}
> $$
>
> **2. Regarding the differentiability of $\\mathcal{N}\_{\\mathcal{D}}$:**
>
> Based on our hard-constraint formulation,$q(x,t) = \\mathcal{P}\_q(x,t) + \\mathcal{D}\_q(x,t) * \\mathcal{H}\_q(x,t)$ (where $q$ is the target physical quantity) and considering the practical implementation of PINNs for PDE training, the model must compute the derivatives of $q(x,t)$ to calculate the PDE loss. The parameters of the $\\mathcal{H}\_q(x,t)$ network ($\\mathcal{N}\_{\\mathcal{H}}$) are updated through backpropagation based on the PDE loss gradients. Therefore, the differentiability of $\\mathcal{D}\_q(x,t)$ has a necessary connection to the backpropagation mechanism. This requirement is also supported by related works such as Lu et al. (2021) [21] and Sun et al. (2020) [11].
>
> We will carefully consider your suggestion to use ablation study results in the appendix to corroborate the necessity of $\\mathcal{N}\_{\\mathcal{D}}$.
>
>
> **For “On A5”:**
>
> Thank you for your suggestion. We will follow your advice and relocate part of our analysis of the mD and mP configurations from the appendix to the main text. In doing so, we will emphasize how the specific structural components of our method contribute to its effectiveness, and highlight the synergistic advantages of the integrated framework compared to partial modifications and existing approaches.

---

> > ### Comment · Reviewer_Zovt · 2025-08-04
> >
> > Ah okay you were talking about the differentiability of $\mathcal{D}_q$ with regard to the inputs $(x,t)$. I was thinking of the differentiability wrt. the parameters, $\theta_\mathcal{D}$.

---

> > > ### Author Response · Authors · 2025-08-04
> > >
> > > Thank you for your clarification and for helping us make this point clearer. We appreciate your kind understanding.

---

### Official Review · Reviewer_BSqT · 2025-07-02

**Clarity:** 2
**Significance:** 2
**Originality:** 2
**Rating:** 4
**Confidence:** 3

**Summary:**

This paper proposes a "Hybrid Boundary Physics-Informed Neural Network" (HB-PINN) to solve Navier-Stokes equations (NSE) with complex boundaries. The method decomposes the solution into a particular solution that satisfies the boundary conditions and a homogeneous solution that satisfies the PDE. This is achieved using a composite architecture of three subnetworks: a "Particular Solution Network" (`N_P`) to approximate the boundary values, a "Distance Metric Network" (`N_D`) to create a boundary-aware mask, and a "Primary Network" (`N_H`) to solve the PDE in the interior. The `N_P` and `N_D` networks are pre-trained and then frozen, while the larger `N_H` network is trained to minimize the PDE residual. The authors test their method on several 2D fluid flow problems, such as flow around a cylinder and flow in an obstructed cavity, and report state-of-the-art accuracy compared to several existing PINN variants.

**Questions:**

1.  **On Originality:** The core idea of solution decomposition (`q = P + D*H`) is a well-established concept for enforcing hard constraints in PINNs. Could you please clarify what you view as the primary conceptual novelty of HB-PINN beyond using neural networks to approximate the `P` and `D` functions, which seems like a direct, albeit complex, implementation of this known principle?
2.  **On Architectural Complexity:** The proposed method uses nine separate DNNs and a multi-stage training process. This is significantly more complex than most standard PINN frameworks. Have you considered a simpler architecture? For example, could a single network be used to output both the solution and an auxiliary distance estimate, or could the `N_P` and `N_D` functions be combined? What is the justification for this high level of architectural complexity?
3.  **On the Necessity of the `N_P` network:** The `N_D` network, when multiplied by `N_H`, already enforces a zero value at the boundary, which is a common way to satisfy homogeneous boundary conditions. The `N_P` network is introduced to handle non-homogeneous boundary values. However, this can also be accomplished with a single term, e.g., `q(x) = BC(x) + D(x)H(x)`, where `BC(x)` is a simple function (or network) that matches the boundary conditions. Could you explain why the separate, pre-trained `N_P` network, which is weakly trained on the PDE, is superior to simpler, more standard ways of enforcing non-homogeneous boundary conditions? A direct comparison to such a simpler baseline would be very informative.

**Ethical Concerns:**

["NO or VERY MINOR ethics concerns only"]

**Final Justification:**

The author addressed most of my concerns, so I have raised my rating. As I am not an expert in this specific field, considering the overall quality of this paper, I still consider it to be borderline. I hope the authors can improve the overall quality of the figures and writing in the future.

**Limitations:**

Yes.

**Paper Formatting Concerns:**

I have not noticed any major formatting issues.

**Quality:**

2

**Strengths And Weaknesses:**

**Strengths:**

1.  **Clear Motivation:** The paper effectively targets the well-known challenge of balancing PDE and boundary losses in PINNs, providing a strong motivation for the proposed method.
2.  **Thorough Experiments:** The method is validated on multiple challenging fluid dynamics benchmarks, with comprehensive comparisons against a wide array of existing PINN variants. The appendices provide extensive details, enhancing reproducibility.

**Weaknesses:**

1.  **Limited Originality:** The core method relies on the classical solution decomposition technique (`q = P + D*H`) to enforce hard boundary constraints, which is a well-established paradigm in the PINN literature. Using neural networks to learn the `P` and `D` components is an implementation detail rather than a fundamental conceptual advance, making the contribution feel incremental.
2.  **Excessive Complexity:** The proposed architecture, involving nine separate networks and a multi-stage, multi-hyperparameter training process, is overly complex and inelegant. This complexity hinders its practicality and scalability compared to simpler hard-constraint methods that achieve similar goals.
3.  **Redundant Components:** The paper fails to convincingly justify the need for both a "Particular Solution Network" (`N_P`) and a "Distance Metric Network" (`N_D`). A single, well-formulated distance function is often sufficient to enforce boundary conditions in similar frameworks. The inclusion of the `N_P` network appears redundant and adds unnecessary complexity without a clear, demonstrated advantage over simpler alternatives.

---

> ### Author Rebuttal · Authors · 2025-07-31
>
> **For Q1**
>
> **A1**: The essential innovation of HB-PINN is not simply employing neural networks to fit the $P$ and $D$ functions, but rather in systematically addressing the fundamental limitations of traditional hard-constraint methods in complex boundary scenarios. The conceptual breakthroughs are:
>
> **1. Breaking the Geometric Adaptability Bottleneck of Traditional Hard Constraints**
>
> Existing hard-constraint methods naively construct $P$ (only guaranteeing boundary point conditions) and $D$ (merely providing geometric distance), decoupling the boundary function $P$ from the solution satisfying the internal physics. While effective for simple boundaries (e.g., regular geometries, low-dimensional spaces), these methods fail to generalize to complex boundaries. This is because existing hard constraints construct $P$ solely considering boundary conditions, resulting in uncontrolled and often irregular output behavior within the spatial interior. This erratic behavior is highly detrimental to the training of the main network when handling complex boundaries.
>
> **2. Proposing a Dual-Network Synergistic Architecture for Complex Boundaries**
>
> **Differentiable Distance Metric Network ($\\mathcal{N}\_{\\mathcal{D}}$)**: Provides a distance function. Used in a power-law form ($\alpha$) to adjust the weight controlled by the distance function within the interior, ensuring the physics (governed by the PDE) dominates away from boundaries.
>
> **Boundary Feature Network ($\\mathcal{N}\_{\\mathcal{P}}$)**: Models the physical feature distribution in the neighborhood of the boundary, rather than simply fitting boundary values. This ensures the output of $\\mathcal{N}\_{\\mathcal{P}}$ is controlled and well-behaved. Furthermore, by incorporating weak constraints derived from the PDE into its construction, $\\mathcal{N}\_{\\mathcal{P}}$ helps reduce the initial training difficulty for the main network when learning the PDE.
>
> Our ablation study (Section C.1 in the Appendix) separately compares the impact of the $P$ and $D$ functions on the results, demonstrating that both the proposed $\\mathcal{N}\_{\\mathcal{P}}$ and $\\mathcal{N}\_{\\mathcal{D}}$ networks play a positive role in solving complex boundary problems. Furthermore, existing research using hard constraints to solve physical models largely remains confined to simple geometric boundaries, with complex boundary problems largely unexplored. HB-PINN was developed precisely to address this unsolved challenge. Comparative evaluations against multiple PINN variants also demonstrate that HB-PINN achieves higher accuracy when handling complex boundary problems.
>
> **For Q2**
>
> **A2**: We understand the concern regarding the complexity of our architecture. However, the HB-PINN method fundamentally consists of three core subnetworks: the main solution network, the boundary condition network, and the distance function network. In our implementation, we define only these three subnetwork frameworks. Under each subnetwork, structurally identical neural networks are used to independently output each physical quantity. This approach is operationally more intuitive for applying constraints to different physical quantities and contributes to maintaining code simplicity and clarity.
>
> Initially, we experimented with a **Multi-output Network** (using a single neural network to simultaneously output all physical quantities to be solved). However, since it uses the same parameter space for computing each physical quantity, differing only at the output layer, this led to a significant reduction in prediction accuracy (see the "Multi-output" section in Table 1).Therefore, we considered decoupling the different physical quantities and employing **Specialized Networks** (each network outputs only one target physical quantity). This setup ensures that the computation of each physical quantity is unaffected by the parameter space of the others. Furthermore, training the partial differential equation inherently requires separately calculating the partial derivatives of different physical quantities. Consequently, this approach actually reduces the operational difficulty during the training process. All PINN methods in our paper use Specialized Networks to ensure the fairness of comparisons.
>
> Moreover, as mentioned in Q1 above, current hard-constraint methods struggle to address the Navier-Stokes equations with complex boundaries (please refer to the results shown in Figures 7 and 9 of our paper), even though they are not necessarily simpler than our method.
>
> The reviewer's suggestion to merge the functionalities of multiple networks is a valid approach for structural optimization. However, integrating them within the hard-constraint architecture remains a significant challenge, representing a classic multi-task optimization problem. Solving this involves difficulties such as: task conflict, compounding and amplification of problem complexity, and challenges in resource allocation. In the context of this work, this manifests because the distance field network $\\mathcal{N}\_{\\mathcal{D}}$ and the boundary solution network $\\mathcal{N}\_{\\mathcal{P}}$ have fundamentally different learning objectives: $\\mathcal{N}\_{\\mathcal{D}}$ needs to accurately fit the geometric topology, while $\\mathcal{N}\_{\\mathcal{P}}$ needs to model the physical characteristics of the boundary layer. We will continue research in this direction, exploring ways to integrate different functionalities to optimize the network structure.
>
>
>
>
> **Table 1:** Performance comparison of network architectures
>
> $$
> \\begin{array}{l l c c}
> \\hline
> \\text{Method} & \\text{Network Structure} & \\text{MSE in Case 2} & \\text{Training time per 1000 epochs (s)} \\\\
> \\hline
> \\text{Multi-output Network} & [6\\times384]+3 & 0.0035 & 138.1 \\\\
> \\text{Specialized Networks} & 3\\times([6\\times128]+1) & 0.0008 & 47.8 \\\\
> \\hline
> \\end{array}
> $$
>
> **Table 2:** Training time comparison of different PINN methods (units: seconds)
>
> $$
> \\begin{array}{lccccccc}
> \\hline
> \\text{Method} & \\text{SPINN} & \\text{hPINN} & \\text{MFN-PINN} & \\text{XPINN} & \\text{SA-PINN} & \\text{PirateNet} & \\text{HB-PINN} \\\\
> \\hline
> \\text{Training time} & 31.1 & 48.3 & 92.5 & 101.5 & 41.4 & 106.9 & 47.8 \\\\
> \\hline
> \\end{array}
> $$
>
> **For Q3**
>
> **A3**: We thank the reviewer for the in-depth discussion of the boundary handling mechanism. We fully understand the theoretical simplicity advantage of the standard formulation $(q = BC(x) + D(x) * H(x))$ you proposed. However, HB-PINN introduces the $\\mathcal{N}\_{\\mathcal{P}}$ network—pre-trained with weak equation constraints instead of a simple $BC(x)$ function—to fundamentally address the challenges inherent in complex boundary scenarios.
>
> Constructing $BC(x)$ directly is feasible when the boundary geometry is regular and the physical distribution is smooth. However, in complex boundary scenarios involving features like multiple obstructions, multiple inlets, or sharp corners, directly constructing a suitable $BC(x)$ becomes extremely difficult or impractical.
>
> For complex boundaries, training the solution network $\\mathcal{N}\_{\\mathcal{P}}$ solely to satisfy boundary conditions leads to distorted or discontinuous outputs within the spatial interior, particularly near junctions between different boundary types. This significantly increases the difficulty of training the main network ($\\mathcal{N}\_{\\mathcal{H}}$), often to the point of preventing convergence.
>
> Specifically, The significant difference between flow field inlet velocity conditions and wall no-slip conditions means that for complex boundaries, forcing the network ($\\mathcal{N}\_{\\mathcal{P}}$) to satisfy both constraints simultaneously can result in uncontrolled behavior in its internal outputs.
>
> As documented in our hPINN experiments within the paper. The main network loss could effectively decrease only after relaxing strict constraints on specific boundary regions (e.g., the wall regions near the inlet:$0 < x < 0.4$, $y=0$ \& $y=1$). Without this relaxation, the main network loss started extremely high and failed to converge, making it impossible to obtain meaningful comparison results. However, this relaxation compromises the satisfaction of the full boundary conditions, which is unacceptable for solving physical models. This precisely illustrates the dilemma of traditional hard-constraint methods with complex boundaries.
>
> The loss progression recorded in Table 3 clearly shows the original hPINN starting with an extremely high loss that failed to decrease normally, while the relaxed hPINN version started with a much more manageable loss value. This is a direct consequence of the uncontrolled $\\mathcal{N}\_{\\mathcal{P}}$ output under complex boundary conditions.
>
> Crucially, by incorporating hard constraints derived from the PDE into $\\mathcal{N}\_{\\mathcal{P}}$'s training (see HB-PINN loss components in Table 3): The initial training loss is significantly reduced and the loss decreases normally during subsequent training.
>
> This demonstrates that incorporating weak PDE constraints into the $\\mathcal{N}\_{\\mathcal{P}}$ network training effectively mitigates irregular behavior in its outputs within the interior domain. Furthermore, because $\\mathcal{N}\_{\\mathcal{P}}$ has undergone preliminary training guided by the governing equation, it substantially reduces the initial training difficulty for the main network ($\\mathcal{N}\_{\\mathcal{H}}$). Therefore, adding weak PDE constraints to the $\\mathcal{N}\_{\\mathcal{P}}$ network training is fundamental and necessary.
>
>
> **Table 3:** Loss comparison across different methods at various epochs
>
> $$
> \\begin{array}{c c c c}
> \\hline
> \\text{Epoch} & \\text{Original hPINN} & \\text{Relaxed hPINN} & \\text{HB-PINN} \\\\
> \\hline
> 0     & 2335810 & 47.44 & 7.129 \\\\
> 5000  & 3318232 & 26.81 & 1.244 \\\\
> 10000  & 5633070 & 25.59 & 1.070 \\\\
> 15000  & 1076081 & 11.20 & 0.810 \\\\
> \\hline
> \\end{array}
> $$

---

> > ### Comment · Reviewer_BSqT · 2025-08-04
> >
> > Thank you for addressing my concerns. I will raise my score to 4.

---

> > > ### Author Response · Authors · 2025-08-04
> > >
> > > We sincerely thank you for your constructive feedback and for raising your score. Your support is greatly appreciated.

---

### Official Review · Reviewer_tZvQ · 2025-07-11

**Clarity:** 3
**Significance:** 3
**Originality:** 3
**Rating:** 5
**Confidence:** 5

**Summary:**

The authors propose Hybrid Boundary PINNs which combines a pretrained boundary condition network along with a primary network which focuses on the inner domain points. A distance metric network is used to enhance the predictions at the boundaries by tuning down the contributions of the pretrained network at the inner points and primary network at the boundary points respectively. This method has been used to solve Navier stokes equations with complex boundaries. Steady state flow past a 2D cylinder, 2D blocked cavity flow with a segmented inlet and a transient flow scenario were considered as examples to demonstrate the efficacy of the proposed framework. The proposed method has been compared against many PINN variants such as soft-constrained PINN (sPINN), hard-constrained PINN (hPINN),  self-adaptive PINN (SA-PINN), modified Fourier network PINN (MFN-PINN), extended-PINN (XPINNs), PirateNets.

**Questions:**

- The model could have outshined if the case studies considered were transient in nature rather than steady state. Most real life use cases of incompressible flows come from unsteady flows involving different vortex dynamical phenomenon like vortex shedding, vortex merging, etc. **Request the authors to consider testing against a transient vortex dominated flow situation.**
- Only a single variant of the distance metric network is considered, radial basis functions, gaussian kernels, indicator functions, etc could also be potential options for these distance metric networks which can be explored to study the difficulty of trainability of PINN model. **Request the authors to comment on the applicability of the other variants and also the superiority of using a power law.**
- While only contours are presented for the predictions, have the authors looked at sectional profiles of velocity and pressure for each case which would be more informative? **Especially provide sectional profiles and temporal history (for transient cases) where the flow-field gradients are strong. Request the authors to provide them for various velocity components and pressure.**
- The ansatz based PINN models for hard constraining is not very new. While the authors have cited Sun et al. (2020)'s work [paper no. 11 in the references] on surrogate modelling in the related work section, they have not highlighted how their distance metric network is different from Sun et al. or compared with their work. But the authors have proposed a distance metric network for hard constraining in the case of complex boundaries which is a relatively novel power law function.
- But the major weakness is that in the study, time evolving problems / unsteady problems are not considered in detail or given equal/higher importance. Unsteady problems if solved, then steady state problems are easier, but the reverse is not true. **Have the authors considered flow past a cylinder at Re > 100 where vortex shedding happens?**
- **Computational cost**: Nowhere is the training vs inference costs in comparison to CFD are presented given that the authors are solving a forward problem. **This is crucial performance related information which needs to be provided comparing with all the other PINN variants as well. Please provide a table detailing the computational cost/ efficiency of the proposed models**

**Ethical Concerns:**

["NO or VERY MINOR ethics concerns only"]

**Final Justification:**

I do not want to change the score since the improvements are only incremental to further enhance the existing paper. The core contribution had already been mentioned in the paper earlier.

**Limitations:**

The key limitation here is that the distance metric function variants need to be evaluated. Especially since the ansatz form for enforcing hard constraints is not new, comparison with existing literature would be useful to frame a stronger case for the proposed method. If not, atleast the limitations of the network would be more clear when such comparison is carried out.

**Paper Formatting Concerns:**

No concerns.

**Quality:**

3

**Strengths And Weaknesses:**

**Strengths**
- The authors have provided a very thorough review of related works and have chosen the latest variants of PINNs relevant to the problem being solved.
- The methodology has been clearly presented without any confusion and nthe details of hyperparameters chosen for each case have also been presented in the appendices without fail.
- Detailed results and discussion are presented in a clear manner with appendices subsequently elaborating and justifying their findings.

**Weaknesses**
- The model could have outshined if the case studies considered were transient in nature rather than steady state. Most real life use cases of incompressible flows come from unsteady flows involving different vortex dynamical phenomenon like vortex shedding, vortex merging, etc.
- Only a single variant of the distance metric network is considered, radial basis functions, gaussian kernels, indicator functions, etc could also be potential options for these distance metric networks which can be explored to study the difficulty of trainability of PINN model.
- While only contours are presented for the predictions, sectional profiles of velocity and pressure for each case would be more informative.
- The ansatz based PINN models for hard constraining is not very new. While the authors have cited Sun et al. (2020)'s work [paper no. 11 in the references] on surrogate modelling in the related work section, they have not highlighted how their distance metric network is different from Sun et al. or compared with their work.

---

> ### Author Rebuttal · Authors · 2025-07-31
>
> **For Q1**
>
> **A1**:We thank the reviewer for their valuable suggestions.
>
> Transient, vortex-dominated flow problems, such as the Kármán vortex street, are indeed classic and important models in fluid mechanics. However, solving these problems within a fully unsupervised learning framework remains a significant challenge for the PINN-based method due to the complex, non-linear interactions between vortices and turbulence. Since our focus is on solving the Navier-Stokes equations with complex boundaries in an unsupervised manner, while the primary challenge in vortex-dominated flow problems is accurately capturing transient turbulent characteristics, these cases are not included in the current version of our paper.
> In response to the request, we tested our method on a transient Kármán vortex street model at $Re ≈ 250$ over $0.1s$ (encompassing two vortex shedding cycles) for further clarification. Both unsupervised and supervised learning settings, using 15 discrete data points within the conical wake region behind the cylinder, were tested.
>
> The MSE calculation results are shown in the table1 below, where the error under unsupervised case is significantly larger compared to the supervised case. Although the MSE in the unsupervised case is still acceptable, it is important to note that, based on our observations, the primary error is concentrated in a small portion of the wake region behind the cylinder, which makes it difficult to accurately capture the Kármán vortex street flow. Although supervised wake predictions are relatively accurate within a few cycles, an accumulation of errors over time remains observable. This underscores that transient problems in long-term sequences present a key area for future research.
> We will consider adding this case study to the appendix for further reference.
>
> **Table 1:** Mean Squared Error (MSE) comparison at different time points
>
> $$
> \\begin{array}{lccccc}
> \\hline
> \\text{Method} & T\\!=\\!0.02 & T\\!=\\!0.04 & T\\!=\\!0.06 & T\\!=\\!0.08 & T\\!=\\!0.10 \\\\
> \\hline
> \\text{Supervised}   & 7.45 \\times 10^{-5} & 1.47 \\times 10^{-4} & 1.93 \\times 10^{-4} & 2.45 \\times 10^{-4} & 3.16 \\times 10^{-4} \\\\
> \\text{Unsupervised} & 1.53 \\times 10^{-3} & 2.12 \\times 10^{-3} & 2.76 \\times 10^{-3} & 3.20 \\times 10^{-3} & 3.55 \\times 10^{-3} \\\\
> \\hline
> \\end{array}
> $$
>
> **For Q2**
>
> **A2**:The radial basis function (RBF), Gaussian kernel function, and indicator function you mentioned are indeed potential alternative approaches for distance metric networks.
>
> **The Gaussian kernel function** is the most commonly used type in **RBF**. An array of Gaussian kernels (or other RBFs) placed along boundaries constructs a signed approximate distance from spatial point x to those boundaries ($s(x) ≈ d(x)$). This approach resembles the R-function method mentioned in our paper, but simultaneously, its construction difficulty and time cost increase with boundary complexity.In contrast, when training distance functions via neural networks, computational cost remains unchanged as long as the network architecture is fixed.
>
> **The indicator function** proves straightforward for precisely locating boundaries or identifying distinct physical subdomains, offering value for multi-region problems. However, its key limitation lies in providing only discrete "in/out of boundary" information, lacking a continuous measure of proximity. Consequently, we consider it less suitable for pure distance metrics and better suited as an auxiliary feature for identifying regional affiliations.
>
> Regarding the **power-law function** we adopted, it requires special clarification: The core function of aforementioned methods like radial basis function (RBF) is to compute the approximate distance from spatial coordinate points to spatiotemporal boundaries. The power-law function, however, serves as a post-processing operation applied after obtaining this approximate distance. When the power exponent exceeds 1, it amplifies weight values in non-zero distance regions, with the amplification effect intensifying proportionally to distance magnitude. Therefore, as a post-processing step, the power-law function does not conflict with the aforementioned distance calculation methods (e.g., RBF) but operates in a complementary and mutually reinforcing manner.
>
>
> **For Q3**
>
> **A3**:Thank you for your valuable suggestion. We will follow this suggestion and include sectional profiles of velocity and pressure, along with the temporal history for the transient cases, in the appendix to provide a more comprehensive understanding of the performance of different approaches, particularly in regions where the flow-field gradients are strong.
>
> **For Q4**
>
> **A4**: We thank the reviewer for highlighting this point and prompting a clearer comparison. The key distinction between our work and that of Sun et al. (2020) lies in the construction of the distance field, which directly relates to the method's capability to handle complex geometric boundaries:
>
> Sun et al.'s work was applied to a vascular model with a relatively simple geometric structure, for which an exact analytical expression for the distance function exists. Consequently, they directly used the analytical solution as the distance field without requiring additional learning.
>
> However, as demonstrated in complex configurations like Case 2 and Case 3 – which are the focus of this paper – analytical distance functions cannot be directly obtained. Computing the distance for such boundaries typically requires computing distances to individual boundary segments and then taking the minimum. Crucially, this minimum operation is non-differentiable, preventing its integration with the main network for joint backpropagation. Therefore, we employ a dedicated neural network (the distance metric network), trained using the minimum distance as labels, to learn a high-precision approximation of the distance field for complex boundaries. This neural network-based fitting approach is universal and can effectively obtain the required (and differentiable) distance function regardless of whether the boundary geometry is simple or complex.
>
>
> **For Q5**
>
> **A5**: We fully recognize the importance of time-evolving (unsteady) problems in fluid mechanics. In fact, they are much more complex than steady problems due to the difficulty of learning non-linear interactions over time. Instead of directly addressing dynamic scenarios with complex boundaries, our goal is to first provide a solution for steady cases and simple transient cases with complex boundaries, which have not yet been adequately addressed in previous methods.
>
> Through our testing of the Kármán vortex street model at $Re ≈ 250$ (as mentioned in A1) and the recorded error profiles (Table 1), we observed that the core difficulty lies in capturing the time-evolving process within the wake region. The HB-PINN method we proposed does not incorporate specific optimizations for this aspect, and therefore currently also struggles to effectively solve such problems. This is a common challenge faced by current unsupervised PINN methods and represents a promising direction for future research.
>
> **For Q6**
>
> **A6**: We fully agree that computational cost is a key metric for assessing the practical value of a method. Table 2 details the comparative training and inference time costs for Case 2 under unified experimental settings (training on an NVIDIA RTX 4090 GPU, CFD run on an Intel Xeon E5-2640 v4 CPU). All PINN methods used the same 20,000 randomly sampled PDE collocation points.
>
> **Training Efficiency**:
> HB-PINN requires only marginally longer training time than the most fundamental sPINN and shows no disadvantage compared to other PINN variants. While maintaining low training costs, HB-PINN achieves significantly lower prediction errors, accomplishing an optimized balance between accuracy and efficiency.
>
> **Inference Efficiency**:
> When predicting results on a $200*200$ grid, the trained PINN method achieves inference times on the millisecond scale, whereas the CFD method requires 17 seconds. This orders-of-magnitude advantage endows HB-PINN with greater potential for real-time simulation, parameter scanning, and optimal control – scenarios demanding high-frequency access to flow field data.
>
> **Table 2:** Comparison of Training and Inference Times
>
> $$
> \\begin{array}{lcc}
> \\hline
> Method & Training\\, Time\\, (1000\\, epochs) & Inference\\, Time \\\\
> \\hline
> \\text{SPINN}       & 31.1\\,\\text{s} & \\ 0.075\\,\\text{s} & \\\\
> \\text{hPINN}       & 48.3\\,\\text{s} & \\ 0.052\\,\\text{s} & \\\\
> \\text{MFN-PINN}    & 92.5\\,\\text{s} & \\ 0.051\\,\\text{s} & \\\\
> \\text{XPINN}       & 101.5\\,\\text{s} & \\ 0.029\\,\\text{s} & \\\\
> \\text{SA-PINN}     & 41.4\\,\\text{s} & \\ 0.034\\,\\text{s} & \\\\
> \\text{PirateNet}   & 106.9\\,\\text{s} & \\ 0.027\\,\\text{s} & \\\\
> \\hline
> \\text{HB-PINN(Ours)}     & \\mathbf{47.8\\,\\text{s}} & \\mathbf{0.042\\,\\text{s}} \\\\
> \\hline
> \\text{CFD}         & \\text{--} & 17\\,\\text{s} \\\\
> \\end{array}
> $$

---

### Note · Authors · 2025-08-14

We sincerely thank the reviewers for their insightful feedback and constructive suggestions, and we also appreciate their recognition of our work. Specifically, Reviewer tZvQ commended the clear methodology and thorough review of related works. BSqT highlighted the strong motivation in addressing the challenge of balancing PDE and boundary losses in PINNs, along with the comprehensive validation on multiple fluid dynamics benchmarks. Reviewer Zovt noted the significance of separately handling boundary and inner regions and the comparisons with multiple baseline methods. Reviewer qJeV emphasized that the results demonstrate potential applicability to a wide range of fluid dynamics problems and other scientific domains, and that the paper represents an effective attempt and progress in this field.

Our main contributions are as follows:

1.Proposes a novel PINN method for solving complex-boundary flow field problems.

2.We introduce a power-function refined distance metric to enhance complex boundary condition handling, thereby significantly improving prediction accuracy.

3.Extensive validations on the Navier-Stokes equations (NSE) under complex boundaries demonstrate superior accuracy over existing PINN methods.

During the rebuttal process, we addressed the critical issues raised, such as:

1.Necessity of Proposed Subnetworks.

We conducted a detailed analysis of the critical roles played by both subnetworks in addressing complex boundary problems, and demonstrated their separate positive impacts on the final results in the ablation study section.

2.Originality Relative to Existing Methods

In the rebuttal, we expounded the difficulties encountered by existing methods in handling complex boundary problems and elucidated the significance of the proposed subnetworks. HB-PINN was developed precisely to address this unsolved challenge.

3.Computational Cost Concerns

We provided additional tabular details demonstrating that the HB-PINN method does not increase computational costs compared to other PINN variants, while achieving higher accuracy when handling complex boundary problems.

We once again thank the reviewers for their constructive suggestions, which are highly valuable for guiding our future work. During the rebuttal process, we carefully considered and addressed all raised questions, gaining the reviewers’ acknowledgment. We will incorporate additional details into the manuscript according to the reviewers’ recommendations.

---

### Decision · Program_Chairs · 2025-09-17

**Decision:**

Accept (poster)

**Comment:**

The paper proposes a novel PINN framework with solid technical contributions and clear motivation. Reviewers were generally positive, highlighting the methodological soundness and empirical results, with only minor concerns about scope and comparisons. The rebuttal provided satisfactory clarifications that strengthened confidence in the work. With overall ratings leaning positive, I recommend acceptance.